



# An evaluation of the LLC4320 global ocean simulation based on the submesoscale structure of modeled sea surface temperature fields

Katharina Gallmeier[1, ⋆], J. Xavier Prochaska[2,3,4,5, ⋆], Peter Cornillon[6, ⋆], Dimitris Menemenlis[7], and Madolyn Kelm[8]

[1]Institute for Defense Analyses, Alexandria, VA, 22305, USA
[2]Affiliate of the Department of Ocean Sciences, University of California, Santa Cruz, CA, 95064, USA
[3]Department of Astronomy and Astrophysics, University of California, Santa Cruz, CA, 95064, USA
[4]Kavli Institute for the Physics and Mathematics of the Universe (Kavli IPMU), 5-1-5 Kashiwanoha, Kashiwa, 277-8583, Japan
[5]Simons Pivot Fellow
[6]Professor Emeritus, Graduate School of Oceanography,University of Rhode Island, Narragansett, RI, 02882, USA
[7]Jet Propulsion Laboratory, California Institute of Technology, Pasadena, CA, 91109, USA
[8]Earth System Science Department, University of California, Irvine, Irvine, CA, 92697, USA
⋆These authors contributed equally to this work.

**Correspondence:** J. Xavier Prochaska (jxp@ucsc.edu)

**Abstract.** We have assembled 2,851,702 nearly cloud-free cutout images (sized $144 \times 144 \, \text{km}^2$) of Sea Surface Temperature (SST) data from the entire 2012–2020 Level-2 Visible Infrared Imaging Radiometer Suite (VIIRS) dataset to perform a quantitative comparison to the ocean model output from the MIT general circulation model (MITgcm). Specifically, we evaluate outputs from the LCC4320 $\frac{1}{48}^{\circ}$ global-ocean simulation for a one-year period starting on November 17, 2011 but oth-

erwise matched in geography and day-of-year to the VIIRS observations. In lieu of simple (e.g., mean, standard deviation) or complex (e.g., power spectrum) statistics, we analyze the cutouts of SST anomalies with an unsupervised Probabilistic AutoEn-coder (PAE) trained to learn the distribution of structures in SST anomaly (SSTa) on ∼10-to-80-km scales (i.e., submesoscale-to-mesoscale). A principal finding is that the LLC4320 simulation reproduces well, over a large fraction of the ocean, the observed distribution of SST patterns, both globally and regionally. Globally, the medians of the structure distributions match

to within $2\sigma$ for 65% of the ocean, despite a modest, latitude-dependent offset. Regionally, the model outputs reproduce mesoscale variations in SSTa patterns revealed by the PAE in the VIIRS data, including subtle features imprinted by variations in bathymetry. We also identify significant differences in the distribution of SSTa patterns in several regions: (1) in the vicinity of the point at which western boundary currents separate from the continental margin, (2) in the Antarctic Circumpolar Current (ACC), especially in the eastern half of the Indian Ocean, and (3) in an equatorial band equatorward of $15°$. It is clear

that (1) is a result of premature separation in the simulated western boundary currents. The model output in (2), the Southern Indian Ocean, tends to predict more structure than observed, perhaps arising from a misrepresentation of the mixed layer or of energy dissipation and stirring in the simulation. The differences in (3), the equatorial band, are also likely due to model errors, perhaps arising from the shortness of the simulation or from the lack of high-frequency/wavenumber atmospheric forcing. Although we do not yet know the exact causes for these model-data SSTa differences, we expect that this type of comparison

will help guide future developments of high-resolution global-ocean simulations.





## 1 Introduction

Ocean General Circulation Models (OGCMs) are an attempt to reproduce the physics and thermodynamics associated with large-scale oceanic processes. The first global implementation of an OGCM with 'realistic' coastlines and bathymetry was undertaken in the early 1970s on a $2° \times 2°$ grid with 12 vertical levels (Cox, 1975). A subjective evaluation of this model's performance compared the dynamic topography 'patterns' of large scale (basin-wide) gyres determined from ship surveys with those obtained from the model. A more quantitative evaluation was also performed by comparing the transport through the Drake Passage determined from hydrographic sections with those obtained from the model—an excellent overview of the early work on OGCMs is provided in K. Bryan's tribute to M. Cox's work (Bryan, 1991). These spatially coarse comparisons made clear that the model reproduced some of the general features of the large-scale circulation but missed others; often those it had missed were off by significant fractions when quantitative comparisons were made. Given the coarse resolution—grid spacing often twice that of the width of major ocean currents such as the Gulf Stream—and the representation for subgrid-scale processes, which attempt to incorporate the physical contribution of processes on scales smaller than the grid spacing, it is not surprising that this model missed some features of large-scale circulation.

In the fifty years since Cox's work, the processing capacity of computers has increased dramatically from $\sim 1$ megaflop, for the Univac 1108 used by Cox, to $\mathcal{O}(10^9)$ megaflops. Likewise, storage capacities have seen similar increases, more efficient codes have been introduced, and the observational data needed to constrain and force the models have seen staggering increases in volume as well as accuracy. Today, the highest-resolution global OGCMs are run on grids ranging from $\frac{1}{12}°$ to $\frac{1}{48}°$ with 100 or more vertical levels (see, e.g., Arbic et al., 2018; Uchida et al., 2022). As a result, these models resolve many mesoscale processes that earlier models had missed. They reproduce quite well most of the large-scale patterns in the global ocean as well as the currents associated with these patterns, offering confidence in studies that use them to predict the evolution of the ocean and atmosphere on a warming planet.

The evaluation methodology described in this manuscript is meant to be applied to unconstrained OGCMs of sufficient resolution to develop vigorous mesoscale and, to some extent, submesoscale variability. At the moment, we lack the observations and estimation tools that are needed to constrain the amplitude and phase of individual mesoscale (and submesoscale) eddies globally and in a dynamically-consistent manner. Therefore, the simulated mesoscale and submesoscale features of free-running models are not expected to match, one-to-one, the observations. While comparing the predicted and modeled fields at an instant in time works for constrained models, the evaluation of free-running models must be performed statistically, since the mesoscale and submesoscale details inevitably differ within the observed field and the one modeled for the same timestamp. Furthermore, to the best of our knowledge, evaluations of the highest-resolution global, free-running OGCMs have, to date, focused on scales substantially larger (one and a half to two orders of magnitude larger) than the horizontal grid spacing of the model (e.g., Fox-Kemper et al., 2019). As such, these evaluations do not assess the capability of such models to reproduce statistically valid measures of the submesoscale structure of their output. The objective of the work presented herein is to





address this deficiency for one of the highest-resolution, global, free-running OGCMs available, specifically, the $\frac{1}{48}^{\circ}$, 90-level

simulation known as LLC4320. This simulation was developed as part of the Estimating the Circulation and Climate of the Ocean (ECCO) project in a collaborative effort between the Massachusetts Institute of Technology (MIT), the Jet Propulsion Laboratory (JPL), and the NASA Ames Research Center (ARC).

To perform the desired evaluation requires a dataset with global coverage spanning at least the LLC4320 time period with a daily cadence (or higher). In addition, it requires observations with spatial sampling comparable to LLC4320 horizontal grid

spacing, which ranges from ∼2 km at the equator to ∼1 km at 70° latitude. Sea Surface Temperature (SST) fields obtained from several different satellite-borne sensors meet these requirements. The selected SST dataset and the LLC4320 simulation are described in more detail in the next section. As discussed in §3, we use an unsupervised machine learning algorithm applied to approximately $150{\times}150\,\mathrm{km}^2$ regions, which we refer to as **cutouts**, to capture a measure of the structure of the SST fields on such scales. By adopting this algorithm, we are intentionally agnostic to specific structures or patterns. The algorithm "learns"

the structures that are dominant in the data and, equally as important, their distribution. Furthermore, it can be applied in the same fashion to observational data and model output. This measure of field structure for the satellite-derived SST fields is then compared statistically with that obtained from similar-sized squares of the model output. The results of these comparisons are discussed in §4.

## 2 Data

### 2.1 Satellite-Derived SST Data

The Visible Infrared Imaging Radiometer Suite (VIIRS) instrument carried on the National Polar-orbiting Partnership (NPP) satellite provided the highest spatial-resolution global SST products, 750 m at nadir (degrading to 1700 m at the swath edge, with at least daily coverage for the period covered by the LLC4320 simulation. VIIRS is a multi-detector instrument for which variations in the gain from detector-to-detector introduces striping in the resulting fields. In addition, geometric distortions

in pixel location arise as the distance from nadir increases, referred to as the bow-tie effect, and render regions more than approximately 500 km from nadir useless for our analysis unless they are corrected. These two issues, striping and inconstant pixel size, significantly impact the structure of the retrieved SST fields. For this reason, we elected to use the National Oceanic and Atmospheric Administration's Level-2P (L2P), $2^{nd}$ full-mission reanalysis (RAN2) of the VIIRS data (Jonasson and Ignatov, 2019), the only product we are aware of that addresses both of these issues [1].

We downloaded all of the RAN2 L2P files for the years 2012–2020, inclusive, from the JPL Physical Oceanography Distributed Active Archive Center (PO.DAAC, https://podaac.jpl.nasa.gov). Each file contains the retrievals from 10 minutes of

---

[1]The Advanced Very High Resolution Radiometer (AVHRR) is not a multi-detector instrument so it does not suffer from the striping and geometric distortion issues associated with data from VIIRS but the coarser spatial resolution, 1.1 km at nadir, introduces greater degradation in resolution with distance from nadir and the noisier instrument results in a product with at least twice the noise than that obtained from VIIRS (Wu et al., 2017). The Sea and Land Surface Temperature Radiometer (SLSTR) instrument carried on the European Sentinel satellites provides an interesting alternative dataset but, given our lack of detailed familiarity with these data, we elected to continue using VIIRS.



satellite data, approximately 5400 scans with 3200 pixels per scan. There are approximately 500,000 files for the period studied. These ∼500,000 files total ∼90 Tb and form the basis of our observational analysis.

## 2.2    SST Output from the LLC4320 Simulation

The LLC4320 simulation was completed in 2015 by coauthor Menemenlis with help from collaborators at MIT and NASA ARC (see, e.g., Rocha et al., 2016a, b; Arbic et al., 2018). The LLC4320 simulation is a global-ocean and sea-ice simulation that represents full-depth ocean processes. The simulation is based on a Latitude/Longitude/polar-Cap (LLC) configuration of the MIT general circulation model (MITgcm; Marshall et al., 1997; Hill et al., 2007). The LLC4320 grid has 13 square tiles with 4320 grid points on each side and 90 vertical levels for a total grid-cell count of $2.2{\times}10^{10}$. Nominal horizontal grid

spacing is ${\frac{1}{48}}^{\circ}$, ranging from 0.75 km near Antarctica to 2.2 km at the Equator, and vertical levels have ∼1-m thickness near surface to better resolve the diurnal cycle. The simulation is initialized from an ECCO, data-constrained, global-ocean and sea-ice solution with nominal ${\frac{1}{6}}^{\circ}$ horizontal grid spacing (Menemenlis et al., 2008). From there, model resolution is gradually increased to LLC1080 (${\frac{1}{12}}^{\circ}$ grid), LLC2160 (${\frac{1}{24}}^{\circ}$ grid), and finally LLC4320 (${\frac{1}{48}}^{\circ}$ grid). Configuration details are similar to those of the ${\frac{1}{6}}^{\circ}$ ECCO solution except that the LLC4320 simulation includes atmospheric pressure and tidal forcing. The

inclusion of tides allows successful shelf-slope dynamics, water mass modification, and their contribution to the global-ocean circulation (Xu et al., 2013). Surface boundary conditions are from the $0.14^{\circ}$ European Centre for Medium-range Weather Forecasting (ECMWF) atmospheric operational model analysis, starting in 2011. Another unique feature of this simulation is that hourly output of full 3-dimensional model prognostic variables were saved, making it a remarkable tool for the study of ocean and air-sea exchange processes and for the simulation of satellite observations. The 0000-GMT and 1200-GMT global

SST fields for the uppermost 1-m level of the LLC4320 output were downloaded using the `xmitgcm` package for the 365-day period starting on November 17, 2011 yielding 730 files totaling ∼0.5 Tb. Hereinbelow, we will use LLC and LLC4320 interchangeably to refer to the ${\frac{1}{48}}^{\circ}$ MITgcm simulation. Although the entire model domain was downloaded, only the region from the southern extreme to 57°N was considered in the geographic analysis to avoid the change in grid geometry occurring at 57°N.

## 3    Methods

### 3.1    Creation of Comparable SSTa Cutouts

Following our previous study on SST patterns (Prochaska et al., 2021), we chose to analyze cutouts, approximately ∼150×150 km² regions extracted from the parent observational and modeled SST fields. The size of these samples was chosen in part to focus on features at scales of ∼30 km or smaller (i.e., submesoscale). Using these modest-sized cutouts also yields a massive number

of cutouts—O($\sim10^6$)—with array dimensions that are easily tractable to machine learning techniques (∼100×100 pixels). In the following subsections, we detail the procedures to generate such cutouts from the VIIRS data and LLC4320 outputs that have nearly equal dimension and geographical coverage.





### 3.1.1 VIIRS Cutouts

In each VIIRS parent field (*image* hereinafter), we identified every $192{\times}192$ pixel$^2$ subarray that has fewer than 2% of its
pixels masked (quality_level $< 5$) because of land, corrupted pixels usually associated with cloud cover, or missing data. This
2% threshold was established after sampling at a wider range of thresholds (as high as $5\%$) and assessing the outputs. For
thresholds greater than $2\%$, we found that clouds significantly bias estimates of the degree of structure obtained by the machine
learning algorithm (discussed in §3.2.1) applied to the datasets of cutouts.

We then divided each image into a grid with cell size of $96{\times}96$ pixel$^2$ and selected the closest cutout to the center of each grid
cell that satisfies the 2% threshold (or 0 if none satisfy). This approach randomized the sampling thus avoiding possible biases
that may have emerged from cutouts sampled on a regular grid. In a well-sampled image, this implies each cutout has $\sim$50%
overlap with its 4 nearest neighbors and $\sim$25% with the next 4 nearest neighbors. Unlike our analysis of MODerate-resolution
Imaging Spectroradiometer (MODIS) data, we did not place a restriction on distance from nadir because the physical size of
the VIIRS pixels varies by approximately a factor of two from nadir to swath edge (Jonasson et al., 2022) compared with a
variation of approximately a factor of five for MODIS.

From the full parent dataset, we extracted 2,851,702 cutouts (limiting to $< 57°$ N; see 3.1.2). The geographical distribution of
these is shown in Fig. 1, which highlights the regions of the ocean that are preferentially cloud free (e.g., the equatorial Pacific
ocean and coastal regions). For this and all subsequent geographic plots, we used the Hierarchical Equal Area isoLatitude
Pixelation (HEALPix)[2] schema (Górski et al., 2005), which tesselates the surface into equal-area curvilinear quadrilaterals and
was introduced for all-sky analysis of astronomical data. Here and throughout the manuscript we adopted nside=64, which
yields a HEALPix cell with approximately $100{\times}100$ km$^2$ area. Values presented are numbers associated with each HEALPix
cell; in this case the number of cutouts in the cell.

For cutouts with one or more masked pixels, we "inpainted" them using the biharmonic algorithm provided in the `scikit-`
`image` software package (van der Walt et al., 2014); the algorithm, we found, performs well even for data with steep gradients
(Prochaska et al., 2021). We then downscaled the arrays with a local mean to $64{\times}64$ pixel$^2$ or approximately $144{\times}144$ km$^2$
at $\sim$2.1 km sampling, which approximately matches the coarsest sampling of the OGCM outputs. Note that the actual size of
cutouts is a function of distance from nadir; we did not resample the observed data to account for these changes. Last, we
demeaned each cutout to produce sea surface temperature anomaly (SSTa) fields. This defines the final, pre-processed dataset
that we use for all VIIRS analyses to follow.

### 3.1.2 LLC4320 Cutouts

Roughly nine years (8 yrs 11 months) of VIIRS data comprise the $\sim$3 million (2,932,452) VIIRS cutouts, whereas the LLC4320
simulation output used for this study spans only one year. To further restrict the VIIRS cutouts to one year would leave too
little data for a full-globe comparison between satellite observations and the model outputs. For this reason, we compare $\sim$9
years of VIIRS data to the one-year model simulation.

---

[2]https://healpix.sourceforge.io

## $Log_{10}$ # Cutouts/HEALPix Cell for VIIRS, LLC Matchups

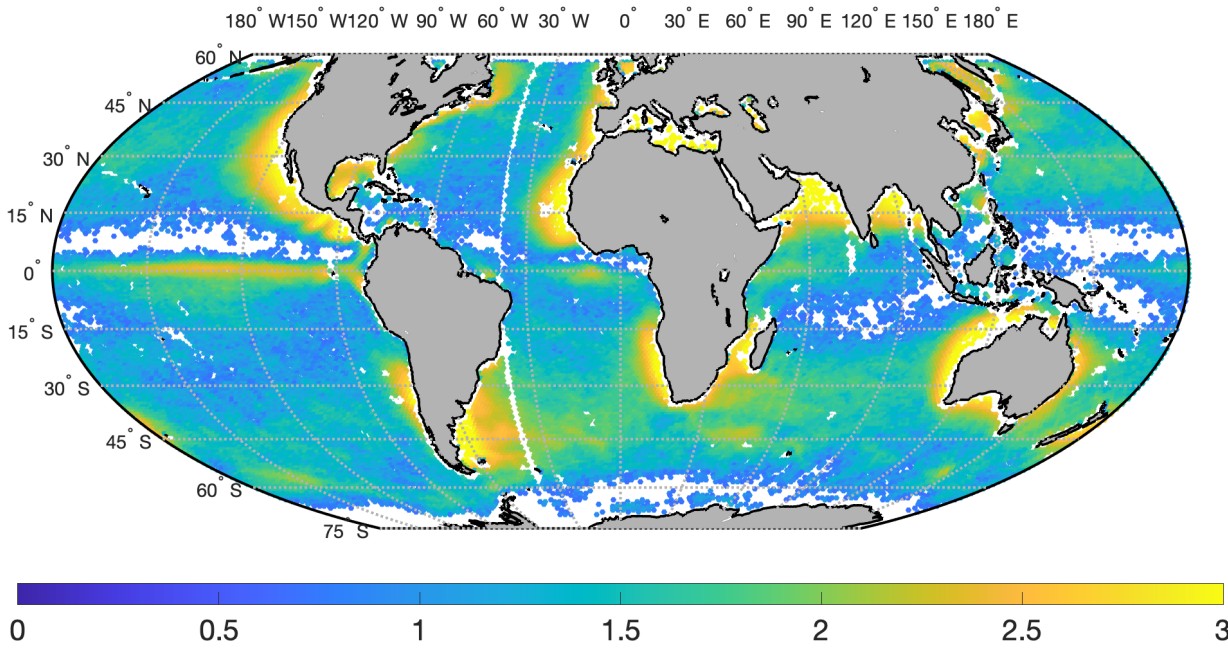

**Figure 1.** Geographic distribution of cutouts in our 98%-clear VIIRS dataset, shown using a $\log_{10}$-based scale of color intensity. Each equal-sized (HEALPix) spatial cell, plotted as dots here, covers approximately 10,000 km$^2$. HEALPix cells with less than five VIIRS cutouts or less than five LLC4320 cutouts are shown in white. Land is shown as light gray. Meridional white line at ∼35°W is due to an LLC4320 sampling artifact.

Our approach to constructing cutouts from the LLC4320 outputs intentionally paralleled the methodology and outputs for VIIRS. We first identified all $64{\times}64\,\mathrm{pixel}^2$ regions that have a valid SST value (i.e., avoiding land). The geographic location (lat, lon) of each of these was recorded. Second, we considered the full year of LLC4320 outputs taken every 12 hours from 2011-11-17 to 2012-11-15 inclusive. For each cutout in the VIIRS sample, we matched in location to the closest valid $64{\times}64\,\mathrm{pixel}^2$ region in the LLC4320 dataset. We then identified the LLC4320 timestamp closest in time from the start of the given year (with the LLC4320 in 12 hour intervals). This is akin to matching on day-of-year and then time-of-day to the nearest

12 hours. This 'climatological' matchup of cutouts was performed to avoid seasonal and regional biases in the sampling.

Much of the analysis presented in subsequent sections of this paper compares the statistics of cutouts in a given HEALPix cell. Ideally, the cutouts associated with a VIIRS-LLC4320 match-up lie in the same HEALPix cell. This, however, is not





always the case; the closest LLC4320 cutout to a VIIRS cutout may lie in an adjacent HEALPix cell when the VIIRS cutout
lies outside of the LLC4320 grid, generally in coast waters.

We wished to maintain an approximately-constant sampling size of $2.25\,\mathrm{km}$ matched to VIIRS or $144{\times}144\,\mathrm{km}^2$ total for
a $64{\times}64\,\mathrm{pixel}^2$ cutout. Therefore, we sized the array extracted from the LLC4320 outputs according to the local size of the
grid, which varies as approximately $\cos(\mathrm{lat})$ for latitudes $\leq 57°\,\mathrm{N}$. At latitudes $> 57°\,\mathrm{N}$, LLC4320 horizontal grid spacing
asymptotes to $\sim 1\,\mathrm{km}$ in the polar cap. To avoid the complications brought by different grid characteristics, we constrained our
analysis to south of $57°\,\mathrm{N}$.

Each extracted LLC4320 array is downscaled to a $64{\times}64\,\mathrm{pixel}^2$ cutout using the local mean. We then injected random noise
using a Gaussian deviate with a standard deviation $\sigma = 0.04\,\mathrm{K}$ based on an analysis of the noise properties of the VIIRS data
(Wu et al., 2017). Lastly, we demeaned each cutout to generate SSTa arrays.

### 3.2 Characterization of the SSTa Cutouts

Each cutout was assigned a Log-Likelihood (LL) metric by the machine learning algorithm, ULMO, used for this work. This
metric describes the frequency of occurrence of the cutout within the full set. The LL metric tends to correlate with the SST
structure, at least at the spatial scales of the fields under consideration here, with structure increasing with decreasing LL
(Prochaska et al., 2021). This simply follows from the fact that the parent sample is dominated by cutouts with little inherent
structure. Comparing the distribution of LL values across the global ocean thus identifies geographic regions where the structure
of the model output at submesoscale-to-mesoscale matches (or fails to match) the observations.

#### 3.2.1 Brief Overview of ULMO

The ULMO machine learning algorithm is a Probabilistic AutoEncoder (PAE; Böhm and Seljak, 2020) designed to assign a
relative probability of occurrence to each cutout in a large dataset. It is an unsupervised method, which learns representations
of the diversity of SSTa patterns without human assessment. The Probabilistic AutoEncoder (PAE) combines two deep learning
algorithms to perform its analysis. The first is an autoencoder that generates a reduced dimensionality representation (aka, a
latent vector) for each cutout in a complex latent space. The second step is a normalizing flow (Papamakarios et al., 2019),
which transforms the autoencoder latent space into a Gaussian manifold with the same dimensionality. One can then calculate
the relative probability of any cutout occurring within the Gaussian manifold with standard statistics. We refer to this relative
probability as the Log-Likelihood (LL) metric.

In the following section, we compare distributions of the LL metric for the VIIRS and LLC4320 cutouts in discrete geo-
graphical regions across the global ocean. This provides a quantitative technique to compare the SST patterns predicted by the
OGCM against those observed in the real ocean. We note that because the LL metric is only a scalar description of a given
pattern's frequency of occurrence, it is possible—in principle—to have similar LL distributions despite qualitative differences
in the SST patterns. This would, however, require a remarkable coincidence, and our visual inspection of regions with con-
sistent LL distributions have not revealed any such examples. We also emphasize that the opposite is not true: regions with
significantly different LL distributions do have qualitatively differing distributions of SST patterns.





Fig. 2 presents galleries of VIIRS and LLC cutouts designed to show how the structure of the cutouts vary as a function of LL. For Fig. 2a the entire LL VIIRS population is divided into quintiles. For each quintile, one VIIRS cutout and one LLC cutout is randomly selected from the 50 LL values nearest to the median of the VIIRS LL distribution. The LL of the median values are shown above the VIIRS cutouts. These galleries show a well defined progression from fields with a large temperature range and accompanying gradients to fields with a smaller temperature range, weaker gradients and less complex patterns.

The correlation between temperature range and LL seen in Fig. 2a was noted by Prochaska et al. (2021) in their discussion of the MODIS dataset. In the analysis to follow (Section 4.2), we compare LL values of VIIRS cutouts with those for LLC cutouts at the same geographic location, which suggests that the temperature range of the cutouts we compare will be similar. (Admittedly, there are a few regions where this is not the case, and we address this when it occurs.) Important from the perspective of the work presented herein is rather how the characteristics of cutouts vary with LL when the temperature range is bound to a small range. To further provide a sense for how cutout characteristics other than temperature range evolve with LL, we present a second gallery of VIIRS and LLC cutouts in Fig. 2b. These cutouts are restricted to have $1 < \Delta T < 1.5\,\mathrm{K}$ where $\Delta T$ is defined as the difference in $90^{th}$ and $10^{th}$ percentiles of the SST distribution: $\Delta T \equiv T_{90} - T_{10}$ where $T_N$ denotes the $N^{th}$-percentile. These galleries were constructed in the same fashion as those for Fig. 2a. In this case, the progression is from cutouts for which SST contours are relatively convoluted to cutouts with relatively straighter contours; in other words, the 'structure' of the cutouts decreases as LL increases. As with the galleries in 2a, the characteristics of the LLC cutouts in this gallery track those of the cutouts in the VIIRS gallery.

### 3.2.2 Training on VIIRS and Evaluation

In Prochaska et al. (2021) we introduced the ULMO algorithm and trained a model with Level-2 (L2) MODIS data sampled at $\sim 2\,\mathrm{km}$ and using $64\times64\,\mathrm{pixel}^2$ cutouts. While this model could have been applied here to the VIIRS and LLC4320 cutouts, we have generated a new ULMO model from the VIIRS dataset. The same hyperparameters derived for ULMO in Prochaska et al. (2021) were adopted here.

We trained the PAE on 150,000 random VIIRS cutouts from 2013 and used the remainder of the data from that year (181,184 cutouts) for evaluation. We trained the autoencoder for 10 epochs with a batch size of 256 and achieved good learning loss convergence. We then trained the normalizing flow for 10 epochs (batch size of 64) and also achieved good convergence. The VIIRS-trained ULMO model was then applied to all of the VIIRS and LLC4320 cutouts to calculate LL for each.

## 4 Results/Discussion

We divide the comparison of the VIIRS SST dataset with the LLC4320 model output, in the context of the patterns learned by ULMO, into those related to the shapes of the two LL probability distributions and those related to their geographic distributions.



## (a) Gallery of the Full Distribution

## (b) Gallery of $\Delta T = [1, 1.5]$ K

**Figure 2.** Galleries of cutouts showing the progression of structure of the associated SST fields as a function of LL. (a) Galleries for the entire set of cutouts. Upper row for VIIRS cutouts, lower row for LLC cutouts. Each cutout is randomly selected from the 50 cutouts with LL nearest to the median VIIRS LL value for the associated LL quintile. The LL of the median is shown above the VIIRS cutout. (b) Similar galleries for all cutouts with $1 < \Delta T < 1.5$ K where $\Delta T \equiv T_{90} - T_{10}$ with $T_N$ denoting the $N^{th}$-percentile.

### 4.1 Overall Statistical Comparison of VIIRS and LLC4320 LL Values

Figure 3 compares the distribution of the LL metric for the 9 years of VIIRS observations against the LLC4320 results matched in space and day-of-year. The two distributions are quite similar, suggesting that, on average, the SST pattern distribution





learned by ULMO from VIIRS is close to the same distribution derived from LLC4320. There are, however, differences, subtle

for much of the range of LL values and not so subtle for $LL > 800$:

- $LL < -375$: This range corresponds to relatively energetic regions, generally associated with strong currents and large SST gradients (Prochaska et al., 2021). In these regions, the probability of finding a VIIRS cutout with a given LL value is lower than that of finding an LLC4320 cutout with the same LL value. This suggests that in dynamic regions, LLC4320 fields tend to have slightly more structure than VIIRS fields, i.e., that the fields are possibly more energetic. An example

of cutouts in this LL range is discussed in the paragraphs following the bolded text **Gulf Stream** in §4.2.2.

- $-375 < LL < 375$: This corresponds to mid-range fields, generally found at mid-latitudes (Fig. 4a), away from eastern and western boundary currents. The probability of finding a VIIRS cutout with a given structure (LL value) in this LL range is increasingly higher, as LL increases from -375 to 375, than that of finding an LLC4320 cutout within this LL range. An example of cutouts in this LL range is discussed in the paragraphs following the bolded text **Southern Ocean**

in §4.2.2.

- $375 < LL < 800$: In this LL range, the probability for the LLC4320 of finding a given LL value is higher—less structure— than for VIIRS. SST fields with these LL values have relatively little structure with retrieval and instrument noise associated with the satellite-derived fields having a relatively larger impact on their observed structure. As noted in §3.1.2, white noise was added to the LLC4320 fields in an attempt to remove the importance of noise in determining the LL

value but, in retrospect, the level of noise may not have been sufficient to address this, hence the higher probability of finding LLC4320 cutouts in this range. Cutouts with $LL > 375$ tend to be found equatorward of approximately $15°$.

- $LL > 800$: The probability distribution for LLC4320 LL values levels off at 800 falling rapidly to zero for values larger than 1100 or so. By contrast, few VIIRS cutouts have LL values greater than 800. This is likely due to noise in the satellite-derived SST cutouts as well as unresolved clouds, discussed in the paragraphs following the bolded text **Equa-**

**torial Band** in §4.2.2.

## 4.2   Geographic Comparison of $\widetilde{LL}_{\mathrm{VIIRS}}$ and $\widetilde{LL}_{\mathrm{LLC}}$ Values

As mentioned in §3.1.1, we compare the geographic distribution of VIIRS LLs with that of LLC4320, based on the HEALPix tesselated surface covering the entire Earth: 41,952 equal-area spatial cells, each of $\sim 100 \times 100\,\mathrm{km}^2$ size. Within each cell, we consider the distribution of LL values from the VIIRS data and LLC4320 output. To minimize the effects of outliers within the

distributions, we utilize the median LL value (designated $\widetilde{LL}$ hereinafter) as a characteristic metric of the structure in SST at a given location. Furthermore, we only consider HEALPix cells containing a least five VIIRS cutouts and five LLC4320 cutouts. The distributions for $\widetilde{LL}_{\mathrm{VIIRS}}$ and $\widetilde{LL}_{\mathrm{LLC}}$ are shown in Fig. 4 and their difference, $\Delta_{LL} = \widetilde{LL}_{\mathrm{VIIRS}} - \widetilde{LL}_{\mathrm{LLC}}$, in Fig. 5. Recall that LL tends to increase with decreasing structure in the SST field, hence positive values of $\Delta_{LL}$ suggest less structure in the satellite-derived cutouts than in the cutouts of the model output and negative values the contrary.



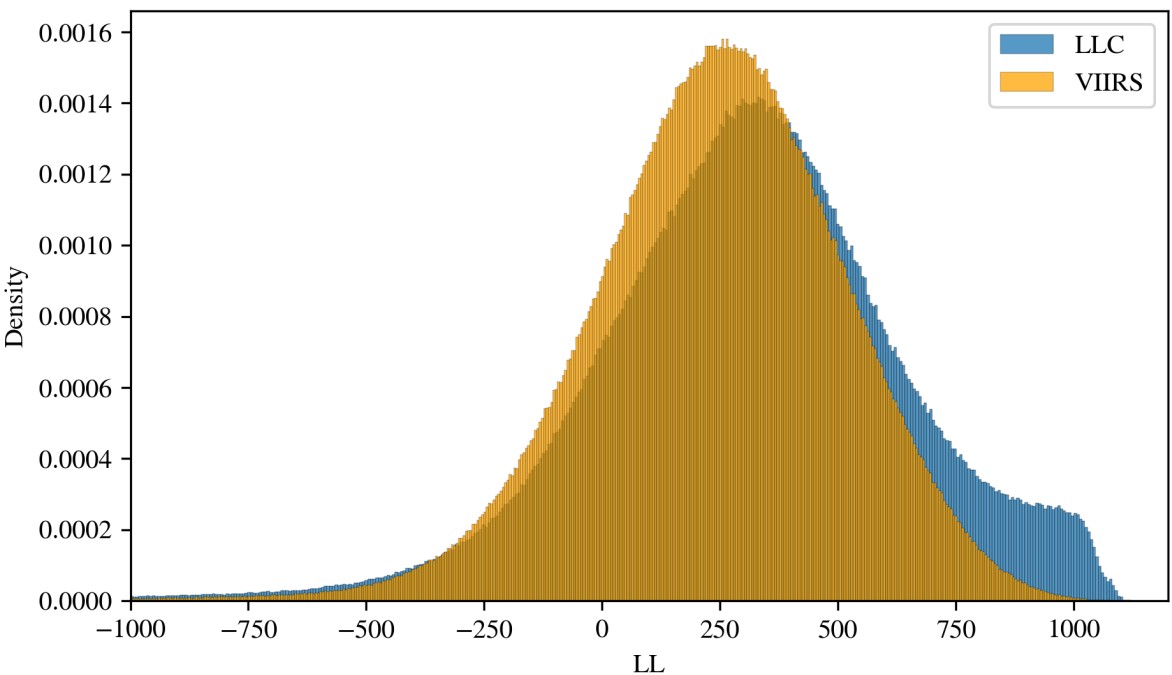

**Figure 3.** Histograms of the LL metric for the full sample of VIIRS and LLC4320 cutouts.

Figure 5 suggests that there are significant differences between the submesoscale-to-mesoscale structure of the simulation and that of the satellite-derived dataset. However, closer examination of the two plots in Fig. 4 suggests that there is significant similarity in the larger scale (> several hundred kilometers) of the two fields; our eyes are drawn to the differences, the large red equatorial regions and blue regions at higher latitudes, not the similarities. We therefore begin by examining the similarities of the fields and then consider their differences.

### 4.2.1 Similarities

To highlight the similarities in the $\widetilde{LL}$ fields on smaller scales ($\mathcal{O}(100\,\mathrm{km})$), we remove from $\widetilde{LL}_{\mathrm{LLC}}$ the large regional differences apparent between them (Fig. 4). This is done by averaging $\widetilde{LL}_{\mathrm{VIIRS}}$ and $\widetilde{LL}_{\mathrm{LLC}}$ over 100 sequential values based on the HEALPix index. Since the vector of HEALPix cells is arranged geographically, the indicies over which the average is performed tend to correspond to a relatively tight geographical region. $\widetilde{LL}_{\mathrm{VIIRS}}$ (black) and $\widetilde{LL}_{\mathrm{LLC}}$ (cyan) are shown in Fig. 6. A 'corrected' $\widetilde{LL}_{\mathrm{LLC}}$, designated as $\widetilde{LL}'_{\mathrm{LLC}}$, is then determined from:

$$\widetilde{LL}'_{LLC_i} = \widetilde{LL}_{LLC_i} - \frac{1}{100}\sum_{j \in B}(\widetilde{LL}_{LLC_j} - \widetilde{LL}_{VIIRS_j})$$





**Figure 4.** (a) HEALPix median LL for VIIRS cutouts ($\widetilde{LL}_{VIIRS}$), (b) $\widetilde{LL}_{LLC}$, (c) Zonal average of $\widetilde{LL}_{VIIRS}$, and (d) Zonal average of $\widetilde{LL}_{LLC}$. Each dot in a and b is associated with an equal area HEALPix cell. White dots correspond to locations with less than five VIIRS cutouts and less than five LLC4320 cutouts in the HEALPix cell. Land is shown in gray.

where $i \in$ all HEALPix cells with at least 5 cutouts in the VIIRS dataset and 5 cutouts in the LLC4320 dataset,

$$B : \left[ \lfloor \frac{i-1}{100} \rfloor * 100 + 1, \lfloor \frac{i-1}{100} \rfloor * 100 + 100 \right]$$

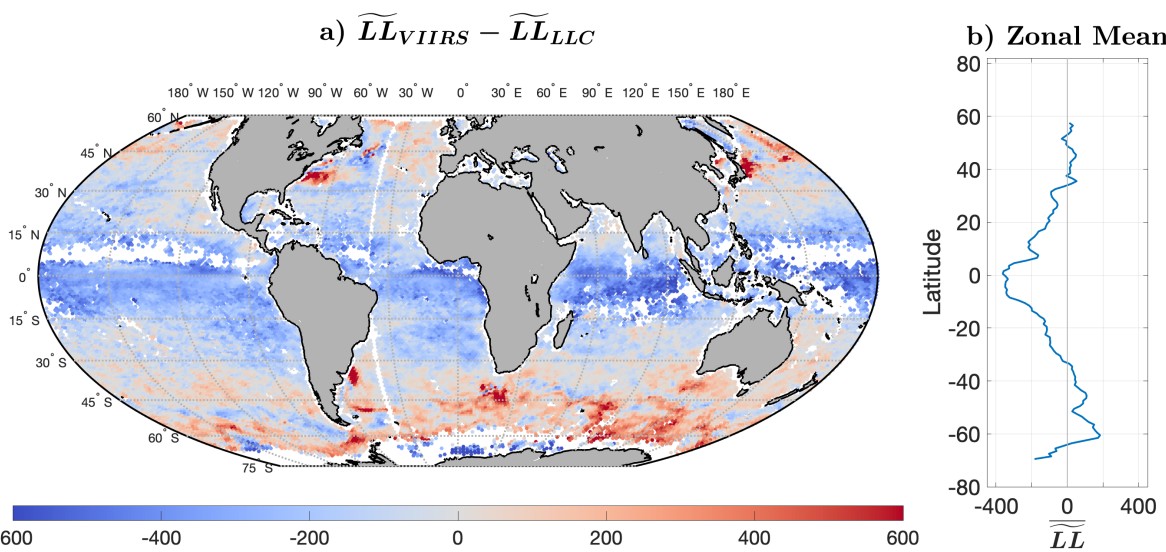

**Figure 5.** (a) $\widetilde{\mathrm{LL}}_{\mathrm{VIIRS}} - \widetilde{\mathrm{LL}}_{\mathrm{LLC}}$. (b) Zonal mean of $\widetilde{\mathrm{LL}}_{\mathrm{VIIRS}} - \widetilde{\mathrm{LL}}_{\mathrm{LLC}}$. It is apparent that the LLC4320 model has SST patterns with less structure in Equatorial regions. In contrast, the dynamic regions of the global ocean (e.g., Western boundary currents) exhibit lower $\widetilde{\mathrm{LL}}_{\mathrm{LLC}}$ indicating a higher degree of structure within these areas in the model output.

and $\lfloor x \rfloor$ dnotes the largest integer less than or equal to x. $\widetilde{\mathrm{LL}}'_{\mathrm{LLC}}$ values are shown in red in Fig. 6.

The $\widetilde{\mathrm{LL}}_{\mathrm{LLC}}$ geographic distribution is shown with that of $\widetilde{\mathrm{LL}}'_{\mathrm{LLC}}$ in Fig. 7. The large scale similarities in the general shape of the distributions is much more evident in this figure. Note also that the zonal mean for $\widetilde{\mathrm{LL}}_{\mathrm{VIIRS}}$ and that for $\widetilde{\mathrm{LL}}'_{\mathrm{LLC}}$ are now quite similar for latitudes equatorward of $60°$.

It is not just the similarities in the large-scale distribution that we find intriguing but also a number of small-scale features. Consider, for example, the $\widetilde{\mathrm{LL}}$ fields at approximately $45°$S in the black and red polygons of Fig. 8. For both $\widetilde{\mathrm{LL}}_{\mathrm{VIIRS}}$ and $\widetilde{\mathrm{LL}}_{\mathrm{LLC}}$ there is a local maximum in $\widetilde{\mathrm{LL}}$ corresponding to a minimum in the structure of the SST cutouts in the black polygon. This feature is associated with a zonal bathymetric ridge at approximately $45°$S crossed by two meridional ridges, one at $44°6'$W and the other at $39°7'$W (Fig. 9b, although the ridges are difficult to see in this figure). The peaks of these ridges are at

depths of approximately $5000\,\mathrm{m}$ in a basin extending to depths of $6000\,\mathrm{m}$. Both LL fields show a band of negative $\widetilde{\mathrm{LL}}$ values to the west and south of the feature. The $\widetilde{\mathrm{LL}}_{\mathrm{VIIRS}}$ field also shows a well-defined band to the north and east while the $\widetilde{\mathrm{LL}}_{\mathrm{LLC}}$ fields only shows a suggestion of such a band but the local minimum is still well-defined in both.

The second feature of interest is the thin, hooked band of low $\widetilde{\mathrm{LL}}_{\mathrm{VIIRS}}$ values—corresponding to relatively more structure in the cutouts—in the red polygon south of South Africa (Fig. 8a). This band is well reproduced in the LLC4320 output (Fig. 8b).

Again, referring to the bathymetric image, (Fig. 9b) it is clear that the shape of this feature is a consequence of the underlying bathymetry. Specifically, it appears that a tendril of the retroflected Agulhas Current has flowed to the south before turning



**Figure 6.** $\widetilde{\mathrm{LL}}$ versus HEALPix cell index: $\widetilde{\mathrm{LL}}_{\mathrm{VIIRS}}$ values are black dots, $\widetilde{\mathrm{LL}}_{\mathrm{LLC}}$ are cyan, and $\widetilde{\mathrm{LL}}'_{\mathrm{LLC}}$ are red.

toward the east to pass through a gap in the southwest-northeast ridge, partially blocking the main part of the retroflected current. The top of the ridge is found at depths of approximately 2000 m while the relatively wide gap, through which the tendril passes, is as deep as 3500 m. The manually digitized center of the feature is shown with the dotted black and solid magenta lines in Fig. 9b; the model appears to reproduce quite accurately this subtle feature in the circulation.




**Figure 7.** As in Fig. 4 except $\widetilde{LL}'_{LLC}$ is shown in (b).

### 4.2.2 Differences

To highlight the regional distributions of significant differences between the structure of satellite-derived SST cutouts and that of SST cutouts produced by LLC4320, we replot in Fig. 10 the data of Fig. 5, masking—dark gray—all values between $-197$ and 197. These thresholds are based on differences between the $\widetilde{LL}$ distribution obtained from the first four years, 2012–2015,

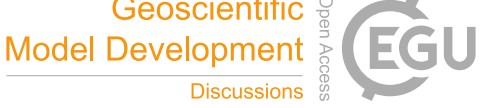

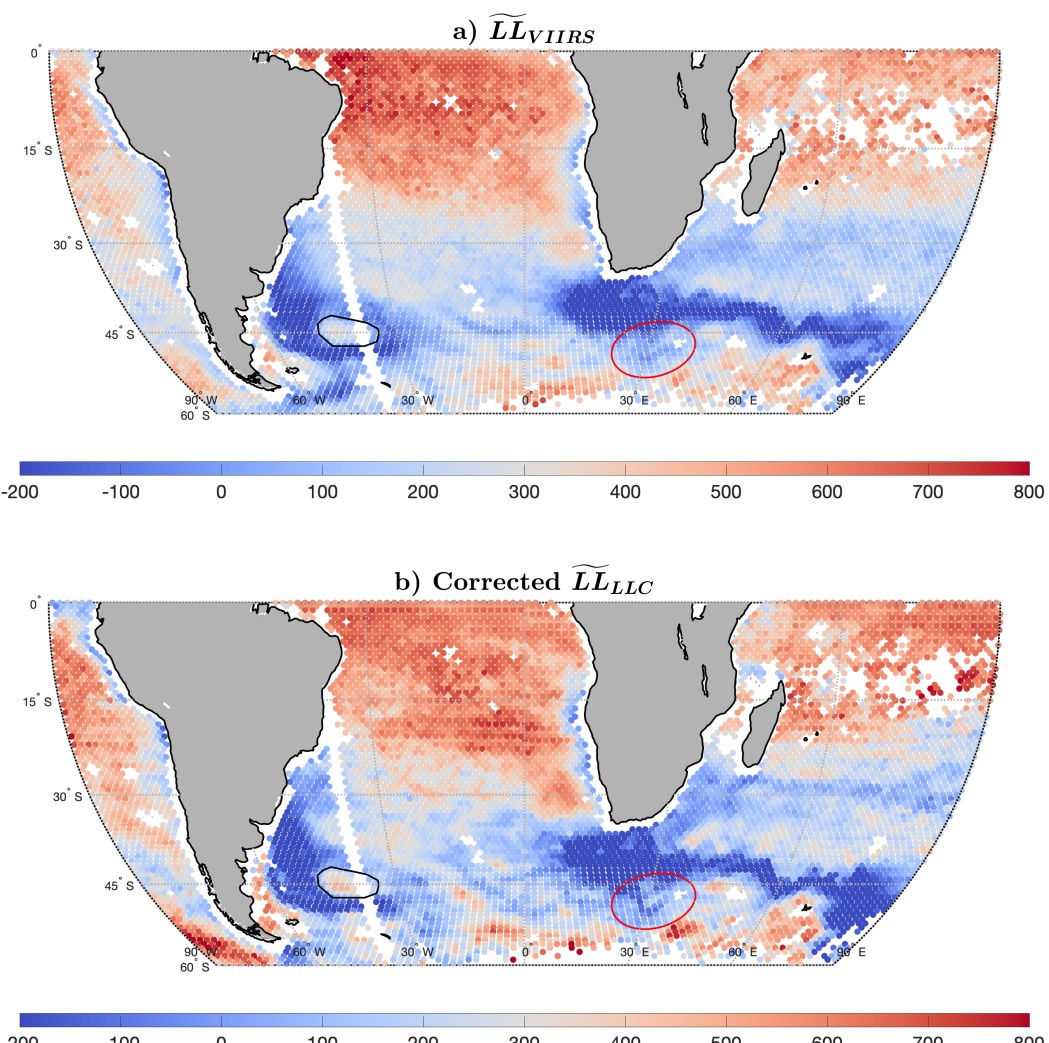

**Figure 8.** As in Fig. 7 but palette constrained to better show features in the two focus area: black polygon at $\sim$35°W, 45°S and red polygon at $\sim$30°E, 50°S.

of the VIIRS dataset and the last four years, 2017–2020, $\widetilde{\mathrm{LL}}_{2012-2015}-\widetilde{\mathrm{LL}}_{2017-2020}$ (see the Appendix). HEALPix cells with $\widetilde{\mathrm{LL}}$ values beyond these thresholds correspond either to regions for which the retrieved VIIRS cutouts are not good measures of the SST or that the LLC4320 output is associated with deficiencies in the simulation.

    $\widetilde{\mathrm{LL}}_{\mathrm{VIIRS}}-\widetilde{\mathrm{LL}}_{\mathrm{LLC}}$ differences evident in Fig. 10 fall into three general groupings: the wide band of negative values (more structure in the VIIRS cutouts than in the LLC4320 cutouts) centered on the Equator, the less continuous band of positive

values in the Southern Ocean, and the very positive (much more LLC4320 structure than VIIRS structure) patches in the

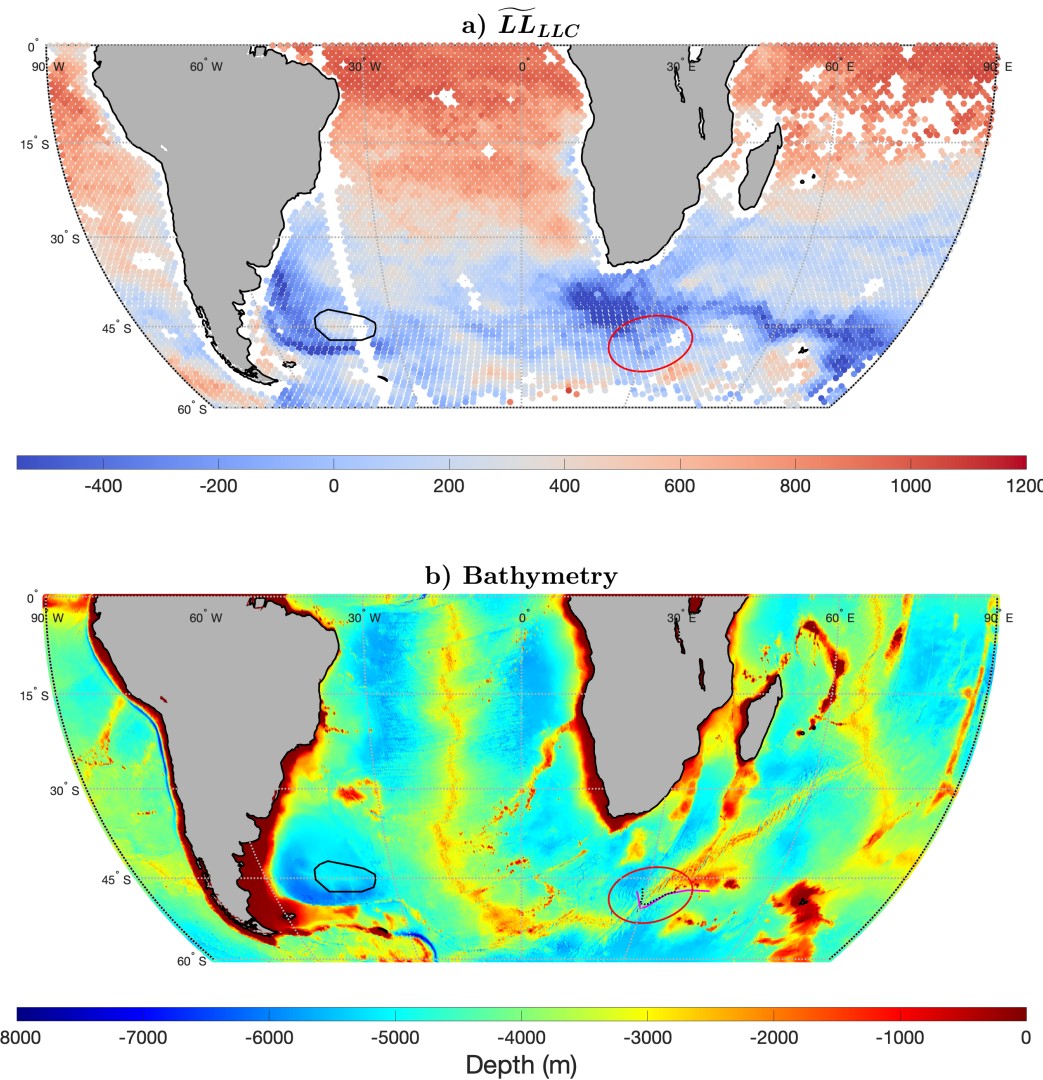

**Figure 9.** (a) Uncorrected $\widetilde{\mathrm{LL}}_{\mathrm{LLC}}$. (b) Bathymetry. Focus regions: black polygon at $\sim35°$W, $45°$S and red polygon at $\sim30°$E, $50°$S. Dotted black and solid magenta lines in the focus area encircled with the red polygon were manually digitized from Fig. 4a and Fig. 9a. for VIIRS and LLC4320, respectively.

vicinity of the separation of western boundary currents from the continental margin, specifically, the Gulf Stream in the western North Atlantic, the Kuroshio in the western North Pacific, the Brazil Current in the South Atlantic, and the Agulhas current where it retroflects south of South Africa. The Equatorial and Southern Ocean bands are also evident in the zonal mean $\widetilde{\mathrm{LL}}_{\mathrm{VIIRS}}-\widetilde{\mathrm{LL}}_{\mathrm{LLC}}$ of the unmasked field shown in Fig. 5b; the zonal mean is roughly flat at -350 from $5°$S to the Equator, rises





rapidly poleward of these two points for about $5°$ of latitude to -150 and then continues to increase approximately linearly, but more slowly, from there ($10°$S and $5°$N) to a value of zero at $30°$N and $30°$S.

In the following we address each of the three regions primarily in the context of SST galleries constructed from two small and geographically close regions (three colored rectangles shown in Fig. 10), which exemplify characteristics of the differences that we find to be of interest.

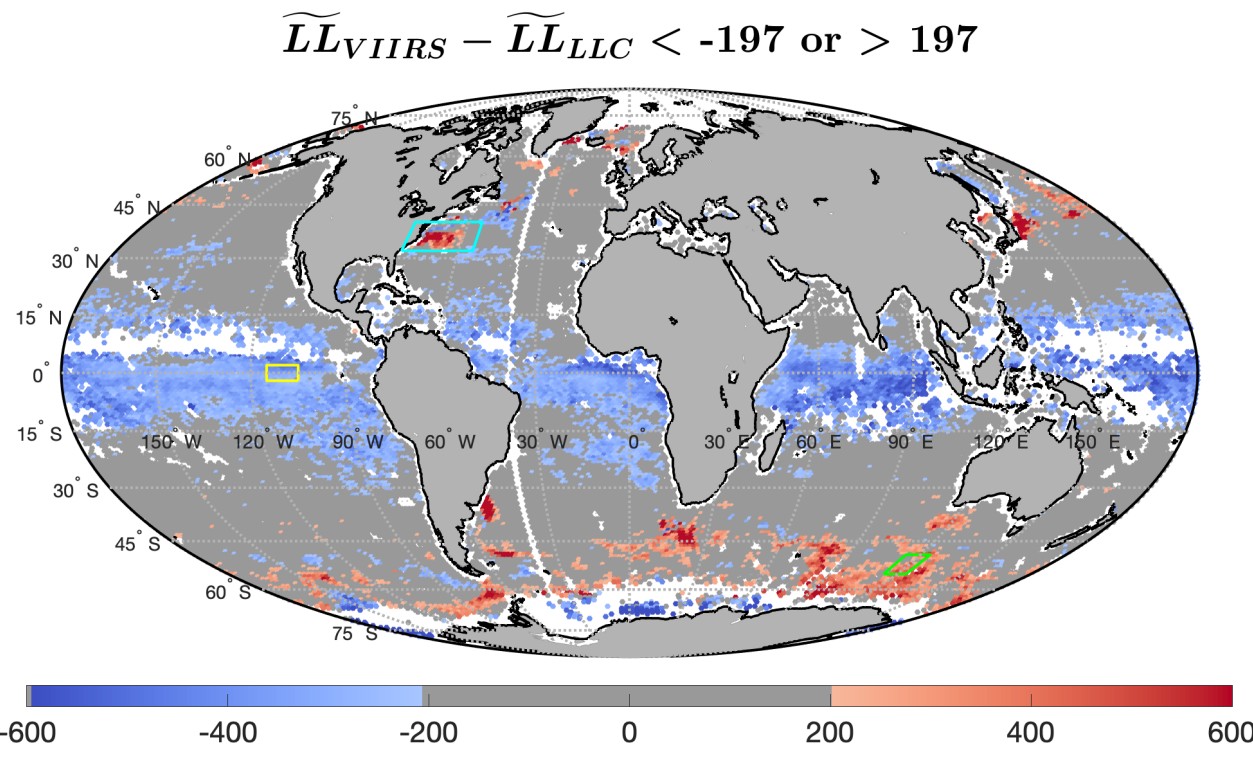

**Figure 10.** As in Fig. 5a with $|\widetilde{LL}_{VIIRS} - \widetilde{LL}_{LLC}| < 197$ masked darker gray to highlight the significant differences between VIIRS observations and the LLC4320 model outputs. The yellow rectangle ($\sim110°$W, $0°$N) designates the focus area for galleries shown in Fig. 13 highlighting Equatorial differences, the green rectangle ($\sim120°$E, $55°$S) for galleries shown in Fig. 15 highlighting Southern Ocean differences and the cyan rectangle ($\sim70°$W, $35°$N) for galleries shown in Fig. 17 highlighting western boundary current differences.

**Equatorial Band:** In general, there are four possibilities for low values of $\widetilde{LL}_{VIIRS} - \widetilde{LL}_{LLC}$ in the equatorial region ($15°$S to $15°$N):





1. *The year simulated, 2012, is atypical, differing from the 2012–2020 mean.* This is very unlikely given the magnitude of the differences, -350 at the Equator, as well as the fact that the zonal mean of $\widetilde{LL}_{2012-2015} - \widetilde{LL}_{2017-2020}$ is essentially flat at 0 equatorward of $50°$ (Fig. A3) suggesting little interannual variability.

2. *Unresolved clouds in the VIIRS cutouts.* Unresolved clouds tend to add structure to the cutouts; the 'quieter' the field, the more significant the impact noise has on the structure, and the greater the decrease will be in $\widetilde{LL}$. To address this potential problem, the 194 cutouts of the HEALPix cell at $36°$N, $112°30'$W were examined for unresolved clouds. Thirty percent were found to be of high quality, 50% were found to be significantly contaminated, and there was uncertainty as to how to classify the remaining 20%. Calculating mean $\widetilde{LL}_{VIIRS} - \widetilde{LL}_{LLC}$ for the high quality only cutouts increased the mean value by 43, small compared to the difference of $-350$ for this HEALPix cell, so this is not likely the primary explanation for the differences. We further address this issue below in the context of galleries of SST fields.

3. *Noise in the VIIRS SST fields.* Cutouts in this part of the ocean tend to have relatively high $\widetilde{LL}$ values (Fig. 4a), corresponding to relatively less structure. This means that noise in the field, which is assumed to change slowly if at all with latitude, will become relatively more important, decreasing $\widetilde{LL}$. Noise has been added to the LLC4320 SST fields in an attempt to address this but possibly not enough resulting in more negative values of $\widetilde{LL}_{VIIRS} - \widetilde{LL}_{LLC}$.

4. *LLC4320 does not reproduce the submesoscale-to-mesoscale structure well in the Equatorial regions.* There is a suggestion based on the examination of a small patch of this region, discussed below, that the model is missing structure in at least some parts of this region.

The rectangular patch [(2°S, 2°N), (105°W, 95°W)] west of the Galapagos Island in the equatorial Pacific (the yellow rectangle in Figs. 10 and 11) stands out because of the significant step of $\widetilde{LL}_{VIIRS} - \widetilde{LL}_{LLC}$ along the Equator. The $\widetilde{LL}$ distribution for the region in the rectangle above the Equator is provided in Fig. 12a and that below the Equator in Fig. 12c. Consistent with the geographic distributions of $\widetilde{LL}$ in Fig. 4, the median $\widetilde{LL}_{LLC}$ value increases from south to north across the Equator while the median $\widetilde{LL}_{VIIRS}$ value decreases, both contributing to a larger structural difference between VIIRS cutouts and LLC4320 cutouts to the north of the equator than that to the south. Histograms of $dT$ are provided because there is a correlation, although weak, between LL and $dT$, while $dT$ is a more readily understood measure of similarity and differences between VIIRS and LLC4320 SST fields. The distribution of $dT_{LLC}$ is similar across the Equator, while that of VIIRS shows a much longer tail consistent with more variability and lower LL values.

Visual examination of the SST fields supports the above conclusions. Consider SST fields of the VIIRS and LLC4320 cutouts in the yellow rectangle of Fig. 11, again separating them into those above the Equator and those below. Galleries of nine cutouts each are shown in Fig. 13. As previously noted each VIIRS cutout was matched in space and day-of-year with an LLC4320 cutout. To generate the galleries shown in Fig. 13, two sets of cutout pairs were formed. One set consisted of 40% of all pairs in the region for which the VIIRS LL values were closest to the median VIIRS LL value for the region. The second set was similarly constructed except based on LLC4320 LL values. The intersection of the two sets defined the pool from which nine cutout pairs were randomly drawn. These pairs for the region above the equator, $0°$ to $2°$N and $105°$ to $95°$W,



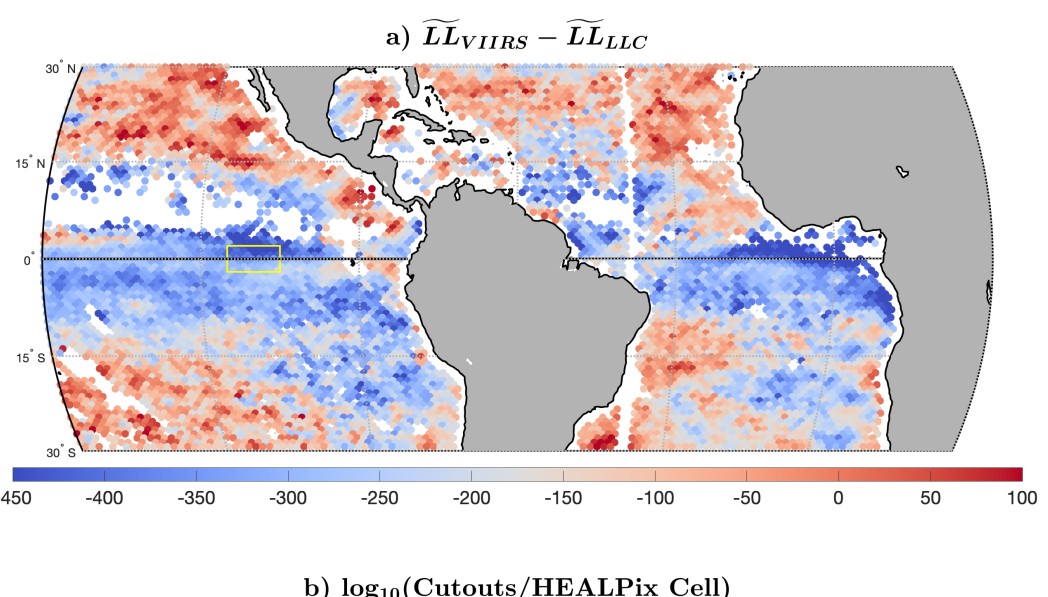

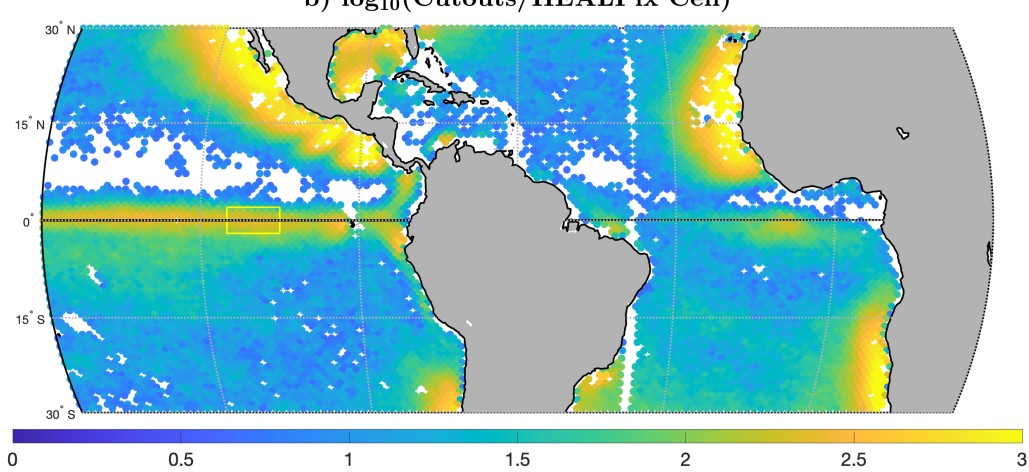

**Figure 11.** (a) $\widetilde{LL}_{\mathrm{VIIRS}} - \widetilde{LL}_{\mathrm{LLC}}$ for the Equatorial Pacific and Atlantic. (b) $log_{10}$ of the # of cutouts/HEALPix cell. Thick horizontal black line is the Equator. Yellow rectangle is the focus area.

are shown in Figs. 13a and b. For each VIIRS cutout in gallery a, its LLC4320 partner is shown in the same location in gallery b. Similarly, Figs. 13c and d show pairs for the region below the Equator. (Remember that because the LLC4320 simulation is free running, there is no reason to expect the features in the pairs to be identical or even similar, it is the 'structure' of the fields that is of interest.) Visually, the SST fields of the LLC4320 cutouts are very similar in both regions showing very little structure with correspondingly large LL values. The VIIRS fields below the Equator show a little more structure than the LLC4320 fields consistent with the smaller LL values. By contrast, most of the VIIRS fields above the Equator have significantly more





**Figure 12.** (a) Histograms of LL for all VIIRS (yellow) and LLC4320 (green) cutouts above the Equator in the focus area. (b) Histograms of $dT$ for the same region as (a). (c) As for (a) except below the Equator. (d) As for (b) except below the Equator. The median values of the distributions are indicated for (a) and (c), as are the number of cutouts contributing to each. The number of cutouts apply to the corresponding $dT$ frames.



structure than any of those in the other three galleries, with correspondingly lower LL values. Also evident in the VIIRS galleries are blemishes in the fields, scattered regions of colder temperatures. We believe these to be unresolved clouds, i.e., clouds not detected by the retrieval algorithm. Because they add structure to the fields, we believe that they decrease the LL value of the cutout. This would reduce the magnitude of $\widetilde{\mathrm{LL}}_{\mathrm{VIIRS}}-\widetilde{\mathrm{LL}}_{\mathrm{LLC}}$ both above and below the Equator but is not likely

to significantly impact the observed $\widetilde{\mathrm{LL}}_{\mathrm{VIIRS(above)}} - \widetilde{\mathrm{LL}}_{\mathrm{VIIRS(below)}}$. Also, note that the distribution of the number of cutouts per HEALPix cell is symmetric about the Equator in the area of interest (Fig. 11b) suggesting that the contribution of clouds to corruption of the LL values is also symmetric about the Equator in the rectangle of interest.

**Southern Ocean:** We examine the same four possibilities for high $\widetilde{\mathrm{LL}}_{\mathrm{VIIRS}}-\widetilde{\mathrm{LL}}_{\mathrm{LLC}}$ values in the Southern Ocean as we did for low values in the Equatorial region:

1. *The year simulated, 2012, is atypical, differing from the 2012–2020 mean.* Possible but unlikely given the extent of the region covered by anomalously high differences and that there were few differences in this region for which $\widetilde{\mathrm{LL}}_{2012-2015} - \widetilde{\mathrm{LL}}_{2017-2020}$ exceeded the $2\sigma$ threshold.

   2. *Unresolved clouds in the VIIRS fields.* Very unlikely because clouds in the VIIRS fields would tend to increase the structure, which would decrease $\widetilde{\mathrm{LL}}$. Hence $\widetilde{\mathrm{LL}}_{\mathrm{VIIRS}}-\widetilde{\mathrm{LL}}_{\mathrm{LLC}}$ would become more positive than if clouds are not present
in the VIIRS fields, rendering the difference more anomalous, not less so.

   3. *Noise in the VIIRS SST fields.* Unlikely because the geophysical variability of SST in these regions tends to overwhelm noise in the VIIRS cutouts. Furthermore, noise in the VIIRS cutouts would tend to reduce the associated $\widetilde{\mathrm{LL}}$ rendering the differences between uncontaminated $\widetilde{\mathrm{LL}}$ VIIRS values and those obtained from the LLC4320 simulation even larger.

   4. *LLC4320 does not reproduce the submesoscale-to-mesoscale structure well in the Southern Ocean.* There is a suggestion
based on the examination of small patches of this region, which we discuss in more detail below, that the model has more structure in this region than is recorded in actual observations. This indicates that the mixed layer or energy dissipation and stirring due to subgrid-scale physics are not represented with sufficient accuracy in these regions.

   In Fig. 14c, we replot the masked $\widetilde{LL}_{VIIRS} - \widetilde{LL}_{LLC}$ field of Fig. 9 for the Southern Ocean south of Australia with $\widetilde{\mathrm{LL}}_{\mathrm{VIIRS}}$ and $\widetilde{\mathrm{LL}}_{\mathrm{LLC}}$ for the same region in the upper two panels. The band of low $\widetilde{\mathrm{LL}}_{\mathrm{VIIRS}}$ and $\widetilde{\mathrm{LL}}_{\mathrm{LLC}}$ values from 45 to 70°E at about
41°S corresponds to the significant structure in the field associated with the Antarctic Circumpolar Current (ACC). (Note that this band originates south of South Africa where the Agulhas retroflection joins the ACC, the band of negative values of $\widetilde{\mathrm{LL}}_{\mathrm{VIIRS}}-\widetilde{\mathrm{LL}}_{\mathrm{LLC}}$ in Fig. 8.) It appears that the ACC as modeled by LLC4320 for 2012 is slightly to the south of the VIIRS ACC for 2012–2020; the positive values of $\widetilde{\mathrm{LL}}_{\mathrm{VIIRS}}-\widetilde{\mathrm{LL}}_{\mathrm{LLC}}$ south of the band and the corresponding negative values to the north of the band. Although a small shift, the fact that the width of the bands for both VIIRS and LLC4320 are virtually
identical suggests that the modeled ACC is a bit to the south of the envelope of paths in this period, i.e., the slight shift may be significant in the context of the modeled processes. Apart from the slight shift to the south, the model appears to have reproduced the current quite well in this region. East of about 70°E $\widetilde{\mathrm{LL}}$ for the modeled ACC is substantially more negative





**Figure 13.** (a) Gallery for VIIRS cutouts above the Equator in the yellow rectangle of Fig. 11a. (b) As in (a) but for LLC4320 cutouts. (c) VIIRS cutouts below the Equator in the yellow rectangle of Fig. 11a. (d) As in (c) but for LLC4320 cutouts. The mean $\widetilde{\mathrm{LL}}$ and $dT$ for all cutouts in the given region (e.g., $0°$ to $2°$N, $105°$ to $95°$W for gallery a) follow the dataset name and the numbers following in parentheses are the mean $\widetilde{\mathrm{LL}}$ and $dT$ of the cutouts in the gallery. The date and time of each VIIRS cutout is shown above it and $\widetilde{\mathrm{LL}}$ and $dT$ for that cutout follow in parentheses. The date of the corresponding LLC4320 cutout (same position in the LLC4320 gallery) is the same as that of the VIIRS cutout. The time is the closest of 0 and 1200 to the VIIRS time. It is evident that the gallery from the VIIRS observations above the Equator shows greater structure than any of the other subsets.





than that of the observed values, resulting in the anomalous $\widetilde{LL}_{\mathrm{VIIRS}}-\widetilde{LL}_{\mathrm{LLC}}$ values in this region. Interestingly, the VIIRS field shows a positive band north and south of the northern branch of the stream as does the LLC4320 field. In fact, the general

pattern of the $\widetilde{LL}_{\mathrm{LLC}}$ has the same general shape as that of $\widetilde{LL}_{\mathrm{VIIRS}}$. Therefore it appears that the model has the correct general structure for the flow in this region but, as we now emphasize, is too energetic.

Galleries of SSTa cutouts in the black rectangles of Fig. 14 (a region for which the model $\widetilde{LL}$ values are in general agreement with the VIIRS values) are shown in Fig. 15a for VIIRS and 15b for LLC4320. Galleries for the anomalous region, the white rectangles in Fig. 14, are shown in Fig. 15c for VIIRS and 15d for LLC4320. Cutouts for these galleries were randomly selected

as described for the generation of galleries for the equatorial region (Fig. 13). The anomalous behavior is clear in the lower two panels; the LLC4320 fields are much bolder with substantially lower $\overline{LL}$ values and larger $dT$ values. The LLC4320 field is clearly more energetic. By contrast, LLC4320 cutouts in the region for which there appears to be agreement are more similar to VIIRS cutouts.

**Gulf Stream:** The conclusions for the four possibilities of high values in the Gulf Stream are similar to those for the Southern

Ocean;

1. *The year simulated, 2012, is atypical, differing from the 2012–2020 mean.* Very unlikely given, as will be shown below, that the modeled Gulf Stream is south of the most extreme southern positions of paths of the Gulf Stream from a number of observational sources.

2. *Unresolved clouds in the VIIRS fields.* Very unlikely because clouds in the VIIRS fields would tend to increase the struc-
ture, i.e., decrease $\widetilde{LL}$. Hence cloud-free fields would tend to increase $\widetilde{LL}_{\mathrm{VIIRS}}-\widetilde{LL}_{\mathrm{LLC}}$, rendering it more anomalous.

3. *Noise in the VIIRS SST fields.* Unlikely because the geophysical variability of SST in these regions overwhelms noise in the VIIRS fields but, if it were to contribute, it would again increase the $\widetilde{LL}_{\mathrm{VIIRS}}-\widetilde{LL}_{\mathrm{LLC}}$, rendering it more anomalous.

4. *LLC4320 does not reproduce the submesoscale-to-mesoscale structure well in the Gulf Stream region.* As will be shown below, this is likely the cause of the differences but, unlike the differences in the Southern Ocean, we believe that these
differences are due to premature separation of the Gulf Stream from the continental margin, i.e., that the Gulf Stream is in the wrong place as opposed to it being in the correct location but too energetic as appears to the case in the ACC south of Australia.

Figure 16a shows $\widetilde{LL}_{\mathrm{VIIRS}}-\widetilde{LL}_{\mathrm{LLC}}$ in the Gulf Stream region downstream of the point at which it separates from the continental margin. Figure 16b shows the same data but masked, showing only the HEALPix cells with values exceeding the

thresholds identified in the Appendix, $\pm 197$. Also shown in these plots is the mean path of the Gulf Stream (magenta line) and its northern and southern extent (black lines). These were determined from manual digitizations of the path of the stream—defined as the maximum cross-stream SST gradient in the vicinity of the stream—in warmest-pixel composites of all AVHRR 1-km SST fields in contiguous 2-day intervals (Lee, 1996). The mean path of the stream was determined by averaging, over all 2-day composites between 1982 and 1999, the point at which these paths intersected integral degrees of longitude. The northern






**Figure 14.** (a) $\widetilde{LL}_{\text{VIIRS}}$ for the focus area of the Southern Ocean, (b) $\widetilde{LL}_{\text{LLC}}$, and (c) masked $\widetilde{LL}_{\text{VIIRS}} - \widetilde{LL}_{\text{LLC}}$. The black rectangle ($\sim$116°W, -50°S) indicates a region of agreement, $\overline{LL_{VIIRS} - LL_{LLC}} = -63$. The white rectangle ($\sim$120°W, -54°S) is an anomalous region with $\overline{LL_{VIIRS} - LL_{LLC}} = 335$.





**Figure 15.** (a) Gallery of 9 randomly selected VIIRS cutouts within the black rectangle of Fig. 14, the region of 'agreement'. (b) Similarly for LLC4320 cutouts in the same region of 'agreement'. (c) VIIRS cutouts in the anomalous region, the white rectangle in Fig. 14. (d) Same as (c) but for LLC4320 cutouts. Dates, times, LL and $dT$ as in Fig. 13.





and southern extents are the latitudes for which 99% of the paths lie to the south—the northern extent—and 99% lie to the
north—southern extent—for 1982–1999. Large positive differences south of the southern extreme suggest that the LLC4320
output contains more structure in its cutouts than VIIRS, whereas VIIRS fields show more structure within the bounds of the
northern and southern extremes.

The large patch of positive values west of $60°$W corresponds to the premature separation of the modeled Gulf Stream from
the continental margin reported by Cornillon and Menemenlis (2018) based on the mean path and extreme envelope of paths
shown in Figs. 16 and on the Ocean Surface Current Analyses Real-time (OSCAR) surface currents (https://podaac.jpl.nasa.gov/
datasetlist, search Keywords: Oceans/Ocean Circulation, Projects: OSCAR) for 2012. Simply put, the modeled Gulf Stream is
found some 250 km to the south of the mean observed stream at $70°$W and approximately 100 km to the south of the southern
extreme observed between 1982 and 1986.

The cause of the positive and negative anomalous values of Fig. 16b become more clear from the individual plots of $\widetilde{LL}_{VIIRS}$
and $\widetilde{LL}_{LLC}$, Fig. 16c and d, respectively. The most negative values of $\widetilde{LL}$ are seen in the satellite-derived fields along the edge
of the continental slope and, in particular, south of Georges Bank, south of the eastern side of the Gulf of Maine and south and
east of the Grand Banks. (Note that the relatively sharp gradient in $\widetilde{LL}$ values follows the 200-m isobath, shown as dotted red
lines in Fig. 16.) Values remain low to the southern extreme of the Gulf Stream. Recall, that these HEALPix values are medians
obtained from all cutouts in the 8+year interval. During this period the Gulf Stream meanders in the envelope with regions of
significant structure at some times and regions of less structure at others, resulting in less structure on the average than is found
near the shelf break in very active regions that are topographically constrained—south of Georges Bank and south and east of
the Grand Banks. The LLC4320 output also shows the most negative values south of Georges Bank but less so south of the
Grand Banks. This is likely associated with the premature separation of the stream from the continental margin, the negative
values of LL between $65°$ and $75°$W and south of about $36°30'N$. Two aspects of interest associated with the model field after
separating are: 1) the relatively smaller width (meridional extent) of the region covered by the Gulf Stream immediately after
separation when compared with the broader distribution associated with the VIIRS data and the rapid increase in LL values—
decrease in structure—at approximately $62°$; the stream appears to die at that point and 2) the positive LL values east of $60°$W
and south of $33°$N, the cause of the statistically significant negative differences when compared with the VIIRS results. The
former may be due to the fact that model simulation is only for one year while the VIIRS data cover 8+ years. The reasons for
the rapid die-off of the stream and the relatively quieter (larger values of LL) east of $62°$ are not obvious.

Next, we examine VIIRS and LLC4320 cutouts inside the Gulf Stream envelope (the white rectangles in Figs. 16c and d
at [($40°$N, $42°$N), ($60°$W, $50°$W)]) and outside the observed Gulf Stream in the region of anomalously low LLC4320 values
associated with the premature separation of the Gulf Stream (the red rectangles in Figs. 16c and d at [($34°$N, $36°$N), ($70°$W,
$60°$W)]). Galleries of nine cutouts each for both VIIRS and LLC4320 for both regions are shown in Fig. 17. Cutouts for
these galleries were randomly selected as described for the generation of galleries for the equatorial region (Fig. 13). The
characteristics of the VIIRS cutouts outside of the Gulf Stream (Fig. 17c) differ substantially from those of the other three
galleries as does the mean LL. VIIRS cutouts within the stream (Fig. 17a) are similar to those in the LLC4320 gallery of
cutouts south of the Gulf Stream (Fig. 17d), consistent the suggestion that the modeled stream separates prematurely from the



a) $\widetilde{LL}_{VIIRS} - \widetilde{LL}_{LLC}$

b) $\widetilde{LL}_{VIIRS} - \widetilde{LL}_{LLC}$

c) $\widetilde{LL}_{VIIRS}$

d) $\widetilde{LL}_{LLC}$

**Figure 16.** (a) $\widetilde{LL}_{VIIRS} - \widetilde{LL}_{LLC}$ for the Gulf Stream region after separation from the continental margin. (b) Masked $\widetilde{LL}_{VIIRS} - \widetilde{LL}_{LLC}$ for same area. (c) $\widetilde{LL}_{VIIRS}$. (d) $\widetilde{LL}_{LLC}$. Thick magenta line shows the mean Gulf Stream path digitized from 2-day AVHRR SST composites for 1982–1999. Thick black lines show the envelope containing 98% of these paths for the same period, 1% beyond the limits on each side of the envelope. Dotted red line is the 200-m isobath. Red and white rectangles show the focus areas from which galleries of cutouts are selected for Fig. 17.

continental margin. The structure of LLC4320 cutouts in the Gulf Stream (Fig. 17b) lies between that of VIIRS cutouts in the stream (Fig. 17a) and LLC4320 cutouts south of the stream (Fig. 17d) as do the LL values. This is consistent with the modeled stream being to the south of the observed stream.





**Figure 17.** (a) Gallery for VIIRS cutouts in red rectangle of Fig. 16c. (b) Same for LLC4320 cutouts in red rectangle of Fig. 16d. (c) VIIRS cutouts in white rectangle of Fig. 16c. (d) LLC4320 cutouts in white rectangle of Fig. 16d. Dates, times, LL and $dT$ as in Fig. 13.





## 5   Conclusions

In this manuscript we set out to confront outputs from the well-adopted LLC4320 simulation with a large dataset of global
observations. Specifically, we have focused on the submesoscale dynamics traced by SST, an observable with decades of global
coverage provided by a series of sensors on remote sensing satellites. This manuscript used L2 data from VIIRS, restricted to
nearly-clear ($\geq 98\%$ cloud free) cutouts with dimension $\sim 150{\times}150\,\mathrm{km}^2$ selected across the ocean.

Our approach for quantitative comparison between data and model is unconventional. We trained a deep-learning Probabilis-
tic AutoEncoder (PAE) on the VIIRS data to learn the distribution of SSTa patterns observed in the ocean and then applied this
PAE to geographically and seasonally matched SSTa cutouts from the LLC4320 model. An advantage of this approach is that
it is intentionally unsupervised; the network learned from the data the features most characteristic of ocean dynamics traced
by SSTa. On the flip side, the results—especially any differences between data and model—are more difficult to interpret. The
LL metric calculated from the PAE is known to correlate with $\Delta T$ and other physical measures of SSTa yet with significant
scatter (Prochaska et al., 2021). And uncertainties are not inherently calculated; instead we have estimated them by applying
ULMO to two independent subsets of VIIRS data.

Proceeding in this manner, we found that, in general, the distribution of SSTa patterns present in the VIIRS observations are
well-predicted by the LLC4320 model (e.g., Fig. 3). Globally, the medians of the LL distributions from VIIRS and LLC4320
agree within $2\sigma$ for 65% of the ocean (Fig. 10). However, there is a modest but significant and latitude-dependent offset between
data and model with the latter exhibiting less structure in the SSTa cutouts near the Equator and greater structure towards
the poles. After correcting for this latitude-dependent offset, we find that the model frequently recovers mesoscale features
imprinted in the LL distributions and seen in the $\widetilde{\mathrm{LL}}_{\mathrm{VIIRS}}$ field. This includes the reproduction of detailed mesoscale dynamics
often forced by deep bathymetric features. We emphasize here that the VIIRS–LLC4320 comparison is being performed on
spatial scales of $\mathcal{O}(50\,\mathrm{km})$ and less and that it is changes in the structure at these scales that is informing the large scale patterns
observed, i.e., the submesoscale structure of cutouts appears to be tied to larger scale processes. One may conclude that the
LLC4320 model has captured salient mesoscale dynamics across the majority of the ocean.

There are, however, a few notable exceptions. One of these is the location of the Gulf Stream, a previously known failure
of the LLC4320 simulation (Cornillon and Menemenlis, 2018). Giving confidence to the approach taken in this manuscript
to evaluate the performance of the LLC4320 simulation is the fact that a known region of concern is clearly identified as
problematic.

A more subtle difference occurs at the Equator. We have shown from the VIIRS data that the structure in the SSTa cutouts
just north of the Equator exceeds that of its southern counterpart (Fig. 13). This difference in SSTa in VIIRS, however, is not
reproduced by the LLC4320 simulation and there is no reason to believe that cross-Equator differences in the VIIRS data are
spurious.

Third, we highlighted inconsistencies between the LLC4320 model and VIIRS observations in the ACC, where the former
frequently exhibits a higher degree of structure in SSTa and, presumably, more energetic surface currents. We attribute this





increased structure to a misrepresentation of the mixed layer and of subgrid-scale processes that are responsible for energy dissipation and stirring.

We hope that our analysis will inspire similar investigations of both global and regional models. With the construction of large, well-curated datasets, as we have done with VIIRS, one may construct a series of tests. The construction of such a dataset
requires intentional decisions on how to extract and preprocess cutouts for direct comparison to model outputs. Furthermore, the data volume $\mathcal{O}(100\,\mathrm{Tb})$ is sufficiently large to require best practices with storage, databasing, and computing. The authors provide their code (including workflow) with the manuscript and encourage discussion with parties interested in building their own similar analyses.

We also wish to emphasize several of the weaknesses of our methodology and identify paths for improvement in future
work. First, and perhaps foremost, we have not accounted for the mismatch in effective spatial resolution between model and observations. The pixelization of the L2 VIIRS product is $\sim$750 m at nadir hence can resolve features in the few kilometer range. The LLC4320 simulation, meanwhile, has a finest cell size of $\sim$1 km but the formulation is not expected to properly resolve features on scales less than $\mathcal{O}(10)$ km (see, e.g., Su et al., 2018). We considered smoothing (i.e., degrading) the VIIRS data to better match the model outputs but were not confident that we could do so with high accuracy. Furthermore, the PAE
itself is effectively smoothing the data by passing each cutout through a 512-dimension bottleneck (e.g., see Fig. 3 of Prochaska et al., 2021).

Related to the above discussion, the initial analysis ignored latitude dependence in SSTa structure, despite the predicted and observed dynamical differences driven by geophysical fluid dynamics. Further work might, for example, vary the size of the cutouts proportional to the Rossby Radius of Deformation. Similarly, if we were to expand the cutout size to larger scales to
better assess mesoscale features, it may become necessary to match the orientation of the data (here dictated by the satellite path) with the model (fixed with rows/columns parallel to longitude/latitude).

Another weakness of our implementation is the lack of any error estimation from the PAE outputs (i.e., for individual LL values). This is a general weakness of deep learning algorithms (but see Bayesian Neural Nets; e.g. Shridhar et al., 2019). Therefore, we approached uncertainty estimation using an empirical estimate generated from subsets of the data (see the
Appendix). While effective, it is approximate and relies on the central limit theorem to assume a Gaussian deviate. Related, the LL metric of ULMO has no intrinsically physical, mathematical or statistical (despite the name) meaning! Future work focused on comparisons of SSTa or other patterns may consider the scattering transform (Mallat, 2012; Cheng and Ménard, 2021), which has sound mathematical underpinning and may allow for proper statistical tests.

Last, but far from least, are the significant "blemishes" in the data that are absent in the model outputs. Foremost are clouds.
The mitigation for clouds adopted here was to (1) limit to cutouts with fewer than 2% of the pixels masked by the retrieval algorithm (Jonasson et al., 2022) and (2) inpaint these masked pixels. The latter step was required for the PAE and is, in general, required for convolutional neural nets, which expect 'complete' fields. In our exploration of the cutouts, however, we identified a high incidence of clouds that were not masked in the VIIRS data. These ranged from minor blemishes in otherwise uniform fields where the clouds generate non-negligible structure (especially evident in Figs. 13a and c) to, in a few cases, corruption
of the entire field in the cutout. Another negative consequence, perhaps the most serious, is the terrific reduction of potential





data and the resultant geographic biases of the dataset that follow from the 98% clear criterion (Fig. 1). To the greatest extent possible, future work must continue to identify and mitigate clouds; our own efforts are well underway.

As we conclude, we emphasize that perhaps the greatest value of this manuscript was the construction and now dissemination of the large dataset of cutouts for comparison with ocean models as well as with other satellite-derived SST datasets. This

includes the software to generate and analyze them. All of these products are publicly available as described below.

*Code and data availability.* All of the data generated and analyzed in this manuscript is publicly available as *parquet* tables and *hdf5* files at Dryad (LINK TO APPEAR). The code developed throughout the project is provided at https://doi.org/10.5281/zenodo.7545904 (doi: 10.5281/zenodo.7545904) or at https://github.com/AI-for-Ocean-Science/ulmo.

### Appendix A: HEALPix Uncertainty

A question, which arises naturally in the context of Fig. 5, is what constitutes a statistically significant difference in $\widetilde{\mathrm{LL}}$ between the model output and the VIIRS fields. To address this, the VIIRS dataset is divided into two 4-year segments, 1 February 2012 through 31 January 2016 (referred to as 2012–2015 hereafter) and 1 January 2017 through 31 December 2020 (2017–2020). Subsequently, $\widetilde{\mathrm{LL}}$ is calculated for each HEALPix cell for each of the two periods. Figure A1 shows the distribution of cutouts for the first of the two periods. Because the periods for which these data are being calculated are substantially shorter than that

of the dataset from which they are drawn (Fig. 1), the number of HEALPix cells with less than 5 cutouts (white in Fig. A1) is substantially larger.

Figure A2 shows a histogram of the differences of the two $\widetilde{\mathrm{LL}}$ fields $\widetilde{\mathrm{LL}}_{2012-2015}-\widetilde{\mathrm{LL}}_{2017-2020}$. Also shown in Fig. A2 is the histogram of the differences of the VIIRS and LLC $\widetilde{\mathrm{LL}}$ fields, $\widetilde{\mathrm{LL}}_{\mathrm{VIIRS}}-\widetilde{\mathrm{LL}}_{\mathrm{LLC}}$. The two vertical black lines denote $\pm 2\sigma$ of the $\widetilde{\mathrm{LL}}_{2012-2015}-\widetilde{\mathrm{LL}}_{2017-2020}$ distribution. We use these in the body of the manuscript to identify significant outliers in

the fields. There are three primary contributors to the variance of the $\widetilde{\mathrm{LL}}_{2012-2015}-\widetilde{\mathrm{LL}}_{2017-2020}$ distribution. First, there is the uncertainty associated with the assignment of an LL value by the machine learning algorithm to each cutout within cell; think of this as instrument noise. Second, cutouts in each cell are being sampled from a three-dimensional space-time region, the spatial extent defined by the cell boundaries (approximately 100 km on a side) and the temporal extent defined by the four-year period from which each distribution is drawn; think of this as the uncertainty of estimated values based on the finite sample

size. Third, difference in the $\widetilde{\mathrm{LL}}$ value between the period covered by the two datasets, i.e., the true geophysical difference. For $\widetilde{\mathrm{LL}}_{2012-2015}-\widetilde{\mathrm{LL}}_{2017-2020}$, the latter is the difference between 2012–2015 and 2017–2020. For $\widetilde{\mathrm{LL}}_{\mathrm{VIIRS}}-\widetilde{\mathrm{LL}}_{\mathrm{LLC}}$ this would be the difference between the simulated period, 2012, and the period from which the VIIRS data is sampled, 2012–2020. This means that the variability of $\widetilde{\mathrm{LL}}_{2012-2015}-\widetilde{\mathrm{LL}}_{2017-2020}$ places an upper bound on uncertainty in the LL values, variability due to position within the cell and the period from which cutouts contributing to the cell are drawn. As shown in the next paragraph,

we believe that the geophysical contribution of uncertainty to the $\widetilde{\mathrm{LL}}_{2012-2015}-\widetilde{\mathrm{LL}}_{2017-2020}$ differences is small hence the variability of $\widetilde{\mathrm{LL}}_{2012-2015}-\widetilde{\mathrm{LL}}_{2017-2020}$ is a good measure of what constitutes a significant deviation between two datasets.





In light of this, we will use two standard deviations of the $\widetilde{LL}_{2012-2015} - \widetilde{LL}_{2017-2020}$ distribution to identify regions in which the model output agrees/disagrees with the satellite-derived fields.

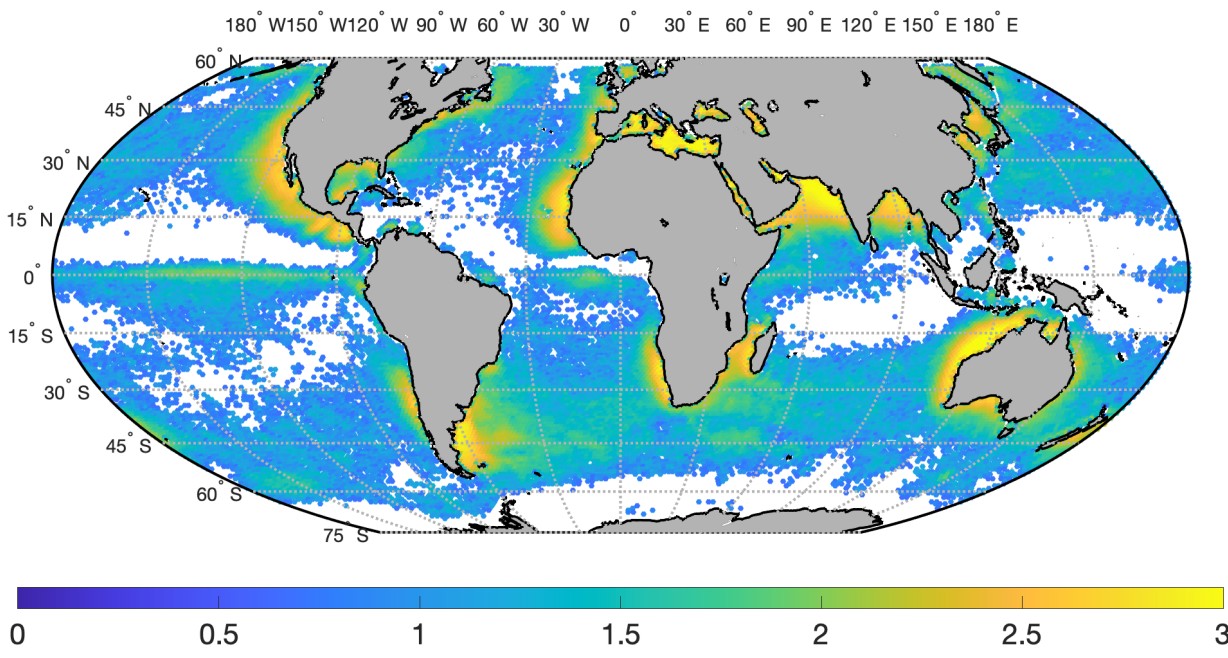

**Figure A1.** Number of VIIRS cutouts per HEALPix cell for the period 2012–2015. HEALPix cells with less than five cutouts for 2012–2015 or less than five cutouts for 2017–2020 are shown in white.

Also of importance in understanding the significance of differences between $\widetilde{LL}_{\text{VIIRS}} - \widetilde{LL}_{\text{LLC}}$ is the degree to which these differences are distributed geographically. Specifically, shifts in major ocean currents as well as changes in forcing from one period to another could result in different structures in the submesoscale-to-mesoscale range, which would display as geographic regions of positive or negative differences between two periods. Figure A3 suggests that, with a few exceptions, the distribution of $\widetilde{LL}_{2012-2015} - \widetilde{LL}_{2017-2020}$ is, in fact, quite random, i.e., at least for this pair of 4-year periods, the differences in submesoscale-to-mesoscale structure is relatively random. There are, however, some regions of more than a few HEALPix cells, which stand out as significantly different between the two periods either in the positive or negative sense. A narrow negative band is evident along the northern edge of the ACC south of the Indian Ocean suggesting that the ACC may have shifted south between 2012–2015 and 2017–2020—more negative values of the difference correspond to less structure in the second period.

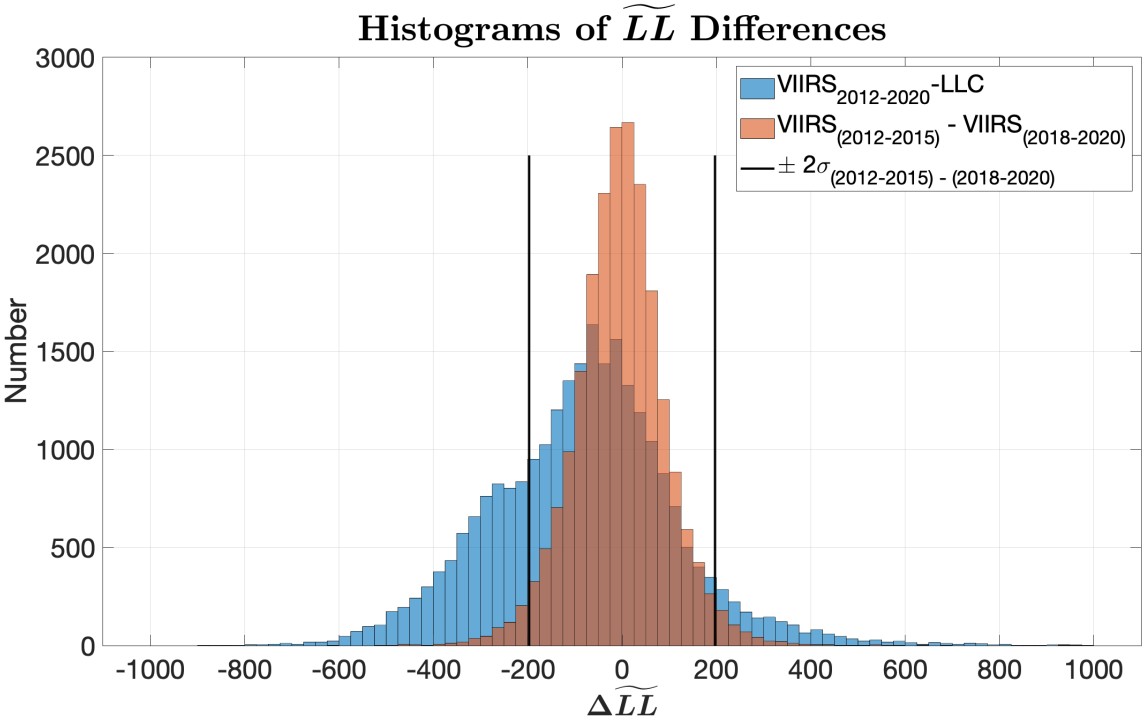

**Figure A2.** Histograms of $\widetilde{LL}_{\mathrm{VIIRS}} - \widetilde{LL}_{\mathrm{LLC}}$ (blue) and $\widetilde{LL}_{2012-2015} - \widetilde{LL}_{2017-2020}$ (light brown). Vertical black lines are $\pm 2\sigma$ of $\widetilde{LL}_{2012-2015} - \widetilde{LL}_{2017-2020}$.

Significant differences are also evident in the vicinity of the Gulf Stream and Kurshio (Fig. A4b masked to show only
HEALPix cells with values more than two standard deviations from the mean, i.e., significant outliers). Figure A4a shows
the $\widetilde{LL}$ distribution for 2012–2015 (the geographic distribution of $\widetilde{LL}$ for 2017–2020 is virtually indistinguishable from that
shown for 2012–2015 in this plot). The dark blue areas on the western side of the North Atlantic and North Pacific north
of approximately 35°N correspond to significant structure in the SST fields in and north of the associated western boundary
currents—the Gulf Stream and Kuroshio, an observation documented in Prochaska et al. (2021). The region of enhanced
differences between the two periods appears to be on the northern edge and to the north of these currents. Figure A5 is a
blow-up of the region in the vicinity of the Gulf Stream: Fig. A5a shows the unmasked $\widetilde{LL}_{2012-2015} - \widetilde{LL}_{2017-2020}$ values
and Fig. A5b shows the masked values. The more positive differences north of the Gulf Stream mean path (magenta line in
the figure) suggest an increase in structure in 2017–2020 compared with that in 2012–2015. That much of the differences are
north of the northernmost extent of the Gulf Stream (upper black line) argues that not only has the mean path of the stream
likely moved to the north in this period but that this displacement resulted in more submesoscale-to-mesoscale turbulence in
the region north of the stream. Of particular interest, although not the focus of this manuscript, is the similarity in the patterns
in the vicinity of the Kuroshio suggesting that the phenomena is hemispheric as opposed to confined to one ocean basin. The





point here is that, although there are some regions of significant differences, these tend to be relatively small and are, in general, associated with strong currents—the Gulf Stream, the Kuroshio, and the ACC.

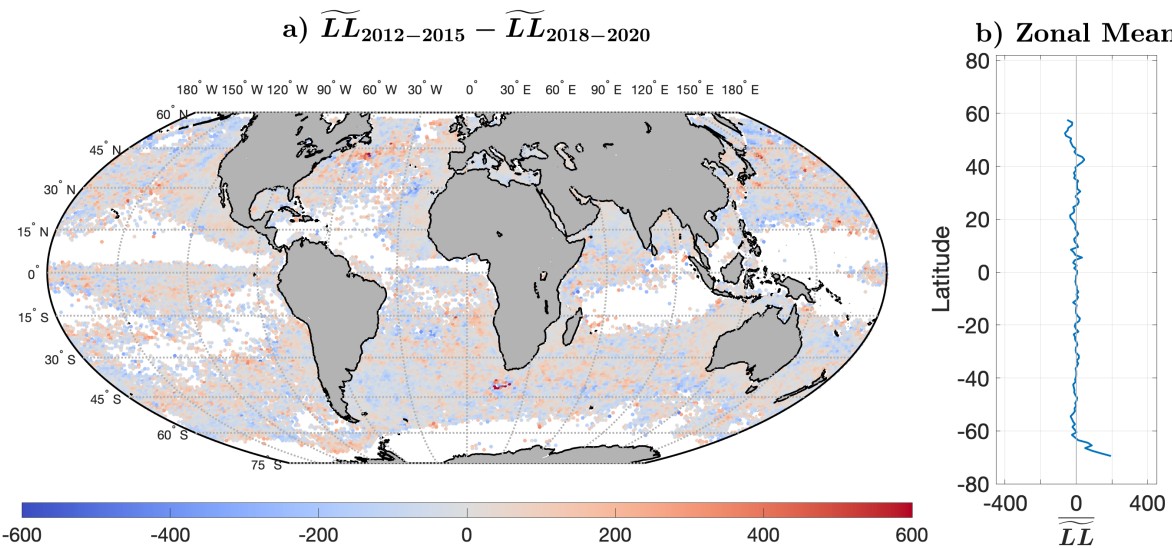

**Figure A3.** (a) $\widetilde{LL}_{2012-2015} - \widetilde{LL}_{2017-2020}$. White areas - less than 5 cutouts in the corresponding HEALPix cell in 2012–2015 and/or 2017–2020. The same color palette is used in this figure as in Fig. 5 to facilitate comparison as well as to emphasize the significant differences in $\widetilde{LL}_{\text{VIIRS}} - \widetilde{LL}_{\text{LLC}}$. (b) Zonal mean of a.

*Author contributions.* KG performed the initial studies on which this manuscript is based. KG, XP and PC contributed to the subsequent analyses of the data. All contributed to the writing of the manuscript, which was led by KG. Figures were generated by KG and PC. XP provided input on the machine learning portion of the project. PC and DM provided input related to physical oceanography. PC provided the satellite data expertise. DM provided the modeling expertise. MK and XP processed the satellite data.

*Competing interests.* The authors declare that they have no conflict of interest.

*Acknowledgements.* JXP acknowledges future support from the Simons Foundation and the University of California, Santa Cruz. DM carried out research at the Jet Propulsion Laboratory, California Institute of Technology, under a contract with NASA, with support from the Physical Oceanography (PO) and Modeling, Analysis, and Prediction (MAP) programs. KG was supported during the summer of 2021 through the University of Rhode Island/Summer Undergraduate Research Fellowship Program in Oceanography, NSF award num-

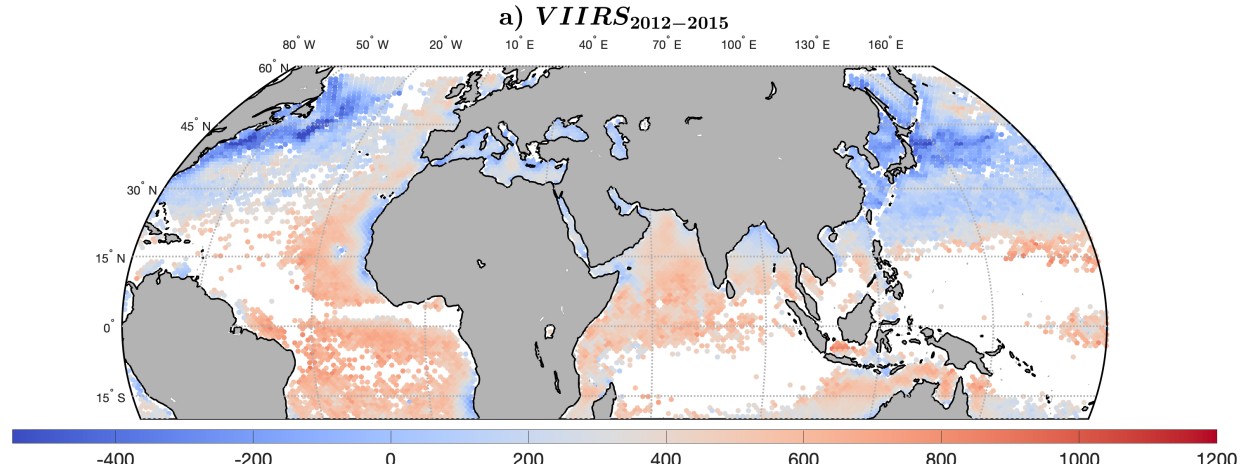

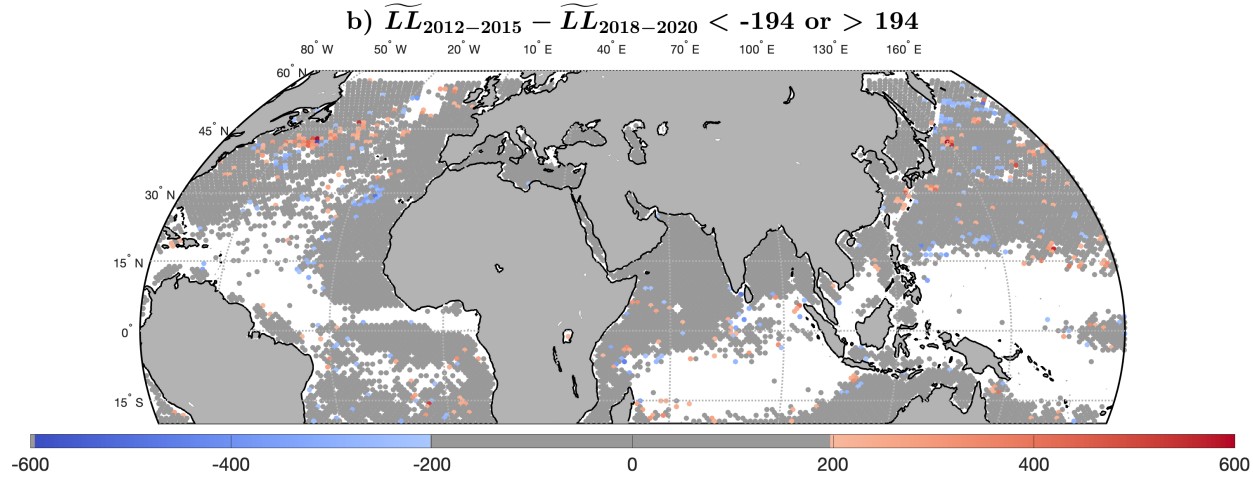

**Figure A4.** a) $\widetilde{\mathrm{LL}}_{2012-2015}$ for the northern hemisphere. b) Masked $\widetilde{\mathrm{LL}}_{2012-2015} - \widetilde{\mathrm{LL}}_{2017-2020}$. Dark gray: $|\widetilde{\mathrm{LL}}_{2012-2015} - \widetilde{\mathrm{LL}}_{2017-2020}| < 2\sigma = 197$. Light gray: land. White - less than 5 cutouts/HEALPix cell for both 2012–2015 and 2017–2020.

ber OCE-1950586. Support for PC was provided by the Office of Naval Research: ONR N00014-17-1-2963, NASA: 80NSSC18K0837,
NASA:80NSSC20K1728 and the State of Rhode Island.

The VIIRS L2 SST data were provided by the Group for High Resolution Sea Surface Temperature and the National Oceanic and Atmospheric Administration and obtained from the National Aeronautics and Space Administration/Physical Oceanography Distributed Active Archive Center. The LLC4320 SST fields were obtained via the xmitgcm package obtained at: https://xmitgcm.readthedocs.io/en/latest/.

Some of the results in this paper have been derived using the HEALPix (K.M. Górski et al., 2005, ApJ, 622, p759) package. The authors
acknowledge use of the Nautilus cloud computing system which is supported by the following US National Science Foundation (NSF)



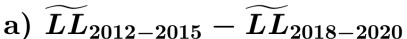

a) $\widetilde{LL}_{2012-2015} - \widetilde{LL}_{2018-2020}$

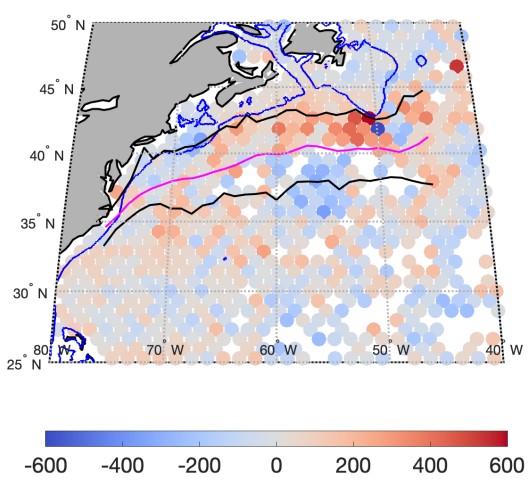

b) $\widetilde{LL}_{2012-2015} - \widetilde{LL}_{2018-2020}$ < -197 or > 197

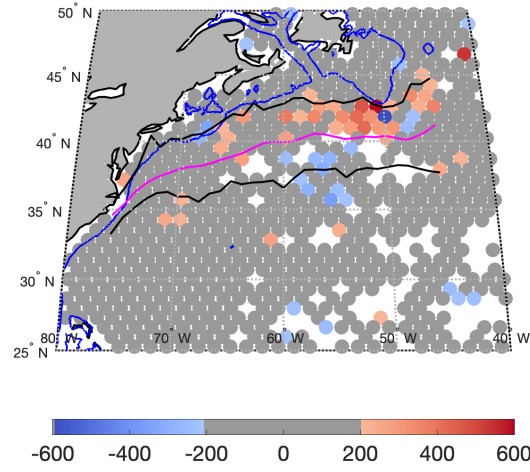

**Figure A5.** (a) As in Figs. A3 and A4 but focused on Gulf Stream region. (b) Masked version of (a). Magenta line between approximate $75°$ W and $45°$ W is the mean path of the Gulf Stream digitized from two-day composites of 1-km AVHRR SST fields for 1982–1999. Black lines are for the northern most and southernmost extents of Gulf Stream paths digitized from the same dataset as (a) but for 1982–1986.

awards: CNS-1456638, CNS-1730158, CNS-2100237, CNS-2120019, ACI-1540112, ACI-1541349, OAC-1826967, OAC-2112167. High-





end computing for the LLC4320 simulation were provided by the NASA Advanced Supercomputing (NAS) Division at the Ames Research Center.





**Acronyms**

| | | |
|---|---|---|
| 590 | **ACC** | Antarctic Circumpolar Current |
| | **AVHRR** | Advanced Very High Resolution Radiometer |
| | **ECCO** | Estimating the Circulation and Climate of the Ocean |
| | **ECMWF** | European Centre for Medium-range Weather Forecasting |
| | **GHRSST** | Group for High Resolution Sea Surface Temperature |
| 595 | **HEALPix** | Hierarchical Equal Area isoLatitude Pixelation |
| | **JPL** | Jet Propulsion Laboratory |
| | **L2** | Level-2 |
| | **L2P** | Level-2P |
| | **LL** | Log-Likelihood |
| 600 | **LLC** | Latitude/Longitude/polar-Cap |
| | **LLC4320** | Latitude/Longitude/polar-Cap4320 |
| | **LLC2160** | LLC2160 |
| | **LLC1080** | LLC1080 |
| | **MITgcm** | MIT general circulation model |
| 605 | **MIT** | Massachusetts Institute of Technology |
| | **MODIS** | MODerate-resolution Imaging Spectroradiometer |
| | **NASA** | National Aeronautics and Space Administration |
| | **NOAA** | National Oceanic and Atmospheric Administration |
| | **NPP** | National Polar-orbiting Partnership |
| 610 | **NSF** | National Science Foundation |
| | **NSF** | National Science Foundation |
| | **OGCM** | Ocean General Circulation Model |
| | **OSCAR** | Ocean Surface Current Analyses Real-time |
| | **PAE** | Probabilistic AutoEncoder |
| 615 | **PO.DAAC** | Physical Oceanography Distributed Active Archive Center |
| | **RAN2** | $2^{nd}$ full-mission reanalysis |
| | **SLSTR** | Sea and Land Surface Temperature Radiometer |





| | | |
|---|---|---|
| | **SST** | Sea Surface Temperature |
| | **SSTa** | sea surface temperature anomaly |
| 620 | **SURFO** | Summer Undergraduate Research Fellowship Program in Oceanography |
| | **URI** | University of Rhode Island |
| | **VIIRS** | Visible Infrared Imaging Radiometer Suite |





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
