# Peer review of "An evaluation of the LLC4320 global ocean simulation based on the submesoscale structure of modeled sea surface temperature fields"

_Geoscientific Model Development, 2023_

## Referee Comment (RC1)

Gallmeier et al. apply a relatively novel statistical method Ulmo to assess a global submesoscale permitting simulation of the ocean (LLC4320). With the increase in computational power, such high-resolution ocean simulations have started to emerge. However, due to issues related to storing and disseminating petabytes of data these simulations produce, a comprehensive assessment on the reality of such simulations has remained a challenge for the ocean modeling community. In this manuscript, the authors compare the fine scale structure in sea-surface temperature (SST) simulated by LLC4320 to satellite observations. They show that while LLC4320 captures the overall basin-scale features in SST, there are discrepancies in the equatorial region and eddy active regions. To my knowledge, this is the first study to attempt to provide a quantitative evaluation of such high-resolution simulations and I believe that their Ulmo method provides a method in evaluating current and future ocean simulations. I recommend their manuscript for publications with minor revisions.

- Lines 46-47: This is more of a comment rather than a criticism but the statement "*the simulated mesoscale and submesoscale features of free- running models are not expected to match, one-to-one, the observations*" is an obvious one. Even if we had the observational tools, free-running models will never match up perfectly to observations due to the chaotic variability of the ocean (e.g. Serazin et al., 2015; Penduff et al., 2018; Leroux et al., 2022; Uchida et al., 2022).

- Lines 223-224 and 375-376: Can the authors comment on how tidal forcing could affect the simulated SST structures that are too energetic? I am asking this because LLC4320 was inadvertently overly forced by tides and consequently is too energetic about the semi-diurnal frequencies (Yu et al., 2019; Arbic et al., 2023).

- Lines 265-272: Is the comparison between $LL_{VIIRS}$ and $LL'_{LLC}$ a fair one given that the latter is corrected by the former and hence the two are no longer independent with each other?

- Lines 285-287: Can the authors explain why the thresholds provide a criteria to judge bad measurements of SST or deficiencies in the simulation?

- Lines 317-318: Could the fact that LLC4320 is a forced ocean simulation and not coupled with the atmosphere be affecting the lack of structure in the equatorial region? In other words, could the atmospheric forcing be damping out the SST structures or could the relatively low temporal resolution in the atmospheric forcing not be exciting sufficient levels of inertial variability (Arbic et al., 2023)? It would be interesting to see if we'd see the same lack of SST structures in the DYAMOND simulation for example, which is air-sea coupled (https://gmao.gsfc.nasa.gov/global_mesoscale/dyamond_phaseII/docs/GEOS-vPICO-EGU21-12782.pdf).

- Line 324 and Figures 13, 15, 17: Is $dT$ equivalent to $\Delta T$ introduced in line 199?

**References**

- Arbic, B. et al., (2023). Frequency dependence and vertical structure of ocean surface kinetic energy from global high-resolution models and surface drifter observations. *ArXiv*.
- Leroux, S. et al., (2022). Ensemble quantification of short-term predictability of the ocean dynamics at a kilometric-scale resolution: a Western Mediterranean test case. *Ocean Sci*.
- Penduff, T. et al., (2018). Chaotic variability of ocean heat content: climate-relevant features and observational implications. *Oceanogr*.
- Serazin, G. et al., (2015). Intrinsic variability of sea level from global ocean simulations: Spatiotemporal scales. *J. Clim*.
- Uchida, T. et al., (2022). An ensemble-based eddy and spectral analysis, with application to the Gulf Stream. *J. Adv. Mod. Earth Sys*.
- Yu, X. et al., (2019). Surface kinetic energy distributions in the global oceans from a high-resolution numerical model and surface drifter observations. *Geophys. Res. Lett*.

---

## Author Response (AR3)

**Addressing comments from Referee 1**

**Comment:**

*Lines 46-47*: This is more of a comment rather than a criticism but the statement "the simulated mesoscale and submesoscale features of free- running models are not expected to match, one-to-one, the observations" is an obvious one. Even if we had the observational tools, free-running models will never match up perfectly to observations due to the chaotic variability of the ocean (e.g. Serazin et al., 2015; Penduff et al., 2018; Leroux et al., 2022; Uchida et al., 2022).

**Author's Response:**

Thank you for listing additional references.

**Changes to Manuscript:**

We revised the wording as suggested and added a reference to Leroux, et al., 2022, which seems to be the most appropriate.
* * *
**Comment:**

*Lines 223-224* and *375-376:* Can the authors comment on how tidal forcing could affect the simulated SST structures that are too energetic? I am asking this because LLC4320 was inadvertently overly forced by tides and consequently is too energetic about the semi-diurnal frequencies (Yu et al., 2019; Arbic et al., 2023).

**Author's Response:**

One possible explanation for more energetic SST fields in LLC4320 relative to VIIRS is the fact that LLC4320 was inadvertently forced with tidal potential that is 10% more energetic than the real ocean (Yu et al., 2019; Arbic et al., 2022) resulting in, for example, more energetic internal tides than observed. A second, in our opinion more likely, explanation is that the 1/48-deg horizontal grid spacing of the simulation, although it is sufficient to enable the gravest modes of mixed layer instabilities to be captured, is insufficient to fully represent the smaller-scale instabilities that would damp the magnitude of the resolved instabilities.

**Changes to Manuscript:**

We added a short paragraph at the end of the Southern Ocean discussion in Section 4.2.2 Differences, starting around line 401.
* * *
**Comment:**

*Lines 265-272:* Is the comparison between *LL* and a fair one given that the *VIIRS LL' LLC* latter is corrected by the former and hence the two are no longer independent with each other?

**Author's Response:**

The reviewer is correct, this is a concern for spatial scales on the same order as the scale used to define the local mean. The magnitude of this spatial scale, order 10,000 km, was not made clear in the manuscript leaving the reader to wonder whether the zonal structure we see in the corrected fields was geophysical or an artifact of the processing. For the features we highlight, the processing did not play a significant role.

**Changes to Manuscript:**

We have added text to explicitly indicate the size of the spatial scale associated with the removal of a background mean.
* * *
**Comment:**

Lines 285-287: Can the authors explain why the thresholds provide a criteria to judge bad measurements of SST or deficiencies in the simulation?

**Author's Response:**

We provided an explanation on the origin of these thresholds in the Appendix. The thresholds are calculated bounds of what log likelihood difference is and what log likelihood difference is not statistically significant enough to indicate a SST structural difference between observational data and model outputs.

**Changes to Manuscript:**

Given the confusion on the reviewer's part, we have decided to add text to the beginning of section 4.2.2 titled "Differences" in an attempt to clarify this.
* * *
**Comment:**

*Lines 317-318*: Could the fact that LLC4320 is a forced ocean simulation and not coupled with the atmosphere be affecting the lack of structure in the equatorial region? In other words, could the atmospheric forcing be damping out the SST structures or could the relatively low temporal

resolution in the atmospheric forcing not be exciting sufficient levels of inertial variability (Arbic et al., 2023)? It would be interesting to see if we'd see the same lack of SST structures in the DYAMOND simulation for example, which is air-sea coupled (https://gmao.gsfc.nasa.gov/global_mesoscale/dyamond_phaseII/docs/GEOS-vPICO -EGU21-12782.pdf).

**Author's Response:**

Agreed, those are possibilities.

**Changes to Manuscript:**

Good point, we have added a paragraph at the end of this section addressing this issue (lines 360-364). Interestingly, we were already planning to do the suggested analysis with Menemenlis' new run.
* * *
**Comment:**

*Line 324* and *Figures 13, 15, 17*: Is $dT$ equivalent to $\Delta T$ introduced in line 199?

**Author's Response:**

Yes, it is. Thank you for catching that.

**Changes to Manuscript:**

Figures and text including $dT$ were changed to $\Delta T$.

**Comment:**

If I may add, could the authors add 2D histograms of LLC plotted against VIIRS and Corrected LLC against VIIRS? I understand that the scatter plot demonstrates that Corrected LLC results in less extrema than LLC but it is difficult to decipher whether there's a bias or not for values between -200 < LL < 600 due to the markers overlapping with each other.

**Author's Response:**

Thank you for the suggestion, we agree the 2D histograms provide additional insight.

**Changes to Manuscript:**

We added a new figure in the revised manuscript and added some text to explain the figure. Please see Figure 7.
* * *
**Comment:**

In terms of scientific reproducibility, it would be better if the authors could document on their Github page (https://github.com/AI-for-Ocean-Science/ulmo) which files/scripts were used in what order to produce their results.

**Author's Response:**

Thank you for your comment!

**Changes to Manuscript:**

We added a ReadMe.md file on the Github page and made minor edits.

**Addressing comments from Referee 2**

**Comment:**

*Title:* after reading the title, my first expression is the important role of submesoscale structure that resolved in the model in the evaluation. However, after reading the abstract, I cannot find any statements that highlight the role of submesoscale. Could the authors make it clearer about this point?

**Author's Response:**

Ulmo was trained to identify submesocale SSTa patterns in cutouts. This was achieved by restricting the cutout size to about 100-100 km^2, thus forcing Ulmo to detect structure on the order of 10-80 km scales. We state this in Section 3.1 Creation of Comparable SSTa cutouts: "The size of these samples was chosen in part to focus on features at scales of ~30 km or smaller." Lines 108-109.

We also include in the introduction a reason for motivation to evaluate based on submesoscale structure, since it has not to our knowledge been done before for global, free-running OGCMs. Lines 52-53.

**Changes to Manuscript:**

We modified the Introduction to emphasize the spatial scales of interest in this analysis.
* * *
**Comment:**

*Line 10-20:* here, the authors introduce the regional differences in the order of Gulf Stream, ACC, and Equatorial regions. But in the main text, the authors actually analyze them in Equatorial regions first, then ACC, and lastly Gulf Stream. To make it consistent, I would suggest the authors change the order here.

**Author's Response:**

Thank you for catching that!

**Changes to Manuscript:**

We changed the regional difference ordering in the introduction to match the manuscript's content.
* * *
**Comment:**

*Line 220*: is it possible to link the LL values to physical dynamic scales here? What does LL = -375 correspond to a spatial scale?

**Author's Response:**

Determining a relationship between LL and dynamical processes is on our agenda. A 1:1 relationship, if one exists, is not obvious. The approach we are taking is to seek a relationship between the full dynamical fields available from the LLC4320 simulation and the associated SST fields. It is possible, however, that there is not a 1:1 relationship for LL. The normalizing flow portion of the ULMO, which allows a straightforward assignment of LL to each cutout, does not require that two cutouts with the same LL are similar in structure, although we believe that in most cases this is likely. We have just submitted a manuscript based on a different ML algorithm (https://arxiv.org/abs/2303.12521) that groups data with similar structures but this has been done in a 2D space thus far rendering it more difficult to assign a single number to each cutout. Having said this we note that this does not invalidate the results of the current manuscript— whatever the relationship is, it applies to both VIIRS fields, on which ULMO was trained and on LLC4320 fields on which the trained algorithm was applied.

Cutouts with a LL of order -375 are generally associated with mesoscale features (see Fig. 17a and d, in this case cutouts in the Gulf Stream) although strong mesoscale features also result in significant submesoscale features as well; i.e., the dominant scale will be mesoscale but there will generally be significant energy at the submesoscale as well.

**Changes to Manuscript:**

We have added a paragraph at the end of Section 3.2.1 summarizing the above.
* * *
**Comment:**

*Line 260*: the equation should be numbered. This is also true for the equations below.

**Author's Response:**

Thank you for catching that.

**Changes to Manuscript:**

Equations are now numbered.
* * *
**Comment:**

*Figure 10:* is there any reason for the authors to choose the two regions to highlight the differences in the Equatorial region and the Southern ocean? It seems that these two are not the regions with the largest differences.

**Author's Response:**

We chose the Equatorial region because we found the very sharp step in LLVIIRS-LLLLC values at the Equator to be intriguing–the step is not close to the Equator, it lies right on it, both in the eastern Equatorial Pacific, the region we highlight, and the eastern Equatorial Atlantic. Our sense is that a feature that is that well defined should help the modelers identify issues in the model giving rise to the differences with the observed fields. We chose the ACC focus area because it appears to be representative of the largest region of disagreement. It would be interesting to look at the anomalous ACC values between 80 and 90E as well but the idea of this manuscript is to introduce the approach and show its application to one model with a few examples.

**Changes to Manuscript:**

We added text to the introduction to clarify that this manuscript's primary purpose is to present a relatively new statistical method in evaluating OGCMs.
* * *
**Comment:**

*Line 315*: Could the authors give more specific explanations about the argument here? What is the possible reason for the model failing to reproduce submesoscale-to-mesoscale structure well?

**Author's Response:**

We believe that the reviewer is referring to what is line 320 in original manuscript version, number 4 in the list under Equatorial Band. We have discussed the possibility (also suggested by referee 1) that this is may be due to coupling between the atmosphere and ocean in this region or, put another way, that the forcing used may not be correct. We hope to address this with the new coupled run that Dimitris Menemenlis is doing.

**Changes to Manuscript:**

Same changes as for comments above. We have added a paragraph at the end of this section addressing this issue (lines 360-364).